**Role of nitrogen and iron biogeochemical cycles on the production and export**
**of dissolved organic matter in agricultural headwater catchments**
Thibault Lambert[1], Rémi Dupas[1,*], Patrick Durand[1]
[1] INRAE, UMR SAS 1069, L'Institut Agro, Rennes, France
* Corresponding author
**Abstract**
To better understand the seasonal variations in environmental conditions regulating
dissolved organic matter (DOM) export in headwater catchments, we combined monitoring of
nitrate, iron, soluble phosphorus and DOM concentration (as dissolved organic carbon; DOC)
and composition (3D fluorescence) in soil and stream waters at regular intervals during one
hydrological year. We installed 17 zero-tension lysimeters in organic-rich top soil horizons
(15 cm below the surface) in the riparian area of a well-monitored agricultural catchment in
French Brittany and collected them at a fortnightly frequency from October 2022 to June
2023. We observed a large increase in DOC concentrations in soil waters during the high
flow period linked to the establishment of Fe-reducing conditions and the subsequent release
of DOM. We also noted that the timing and the spatial variability in Fe(II) biodissolution in
soils was regulated by nitrate from agricultural origin and the heterogeneity of water flow
paths at the hillslope scale. Contrary to our current understanding of DOM export in
headwater catchments, these results lead us to consider the winter high flow period as an
active phase of both DOM production and export.
**1. Introduction**
Dissolved organic matter (DOM) is a key component of the ecological and biogeochemical
functioning of aquatic ecosystems (Hanson et al., 2015), affecting for instance light
penetration (Kelly et al., 2001), pollutant transport (Aiken et al., 2011), aquatic microbial
metabolism (Wetzel, 1992), and the treatment of drinking waters (Chow et al., 2005). Aquatic
DOM, which is mainly of terrestrial origin, represents a fundamental link between the
terrestrial, oceanic, and atmospheric compartments of the global carbon cycle (Dean et al.,
2020; Battin et al., 2008). Unravelling the sources and drivers of DOM export has become an
urgent environmental issue in a context of long-term increasing concentrations of dissolved
organic carbon (DOC, a proxy for DOM content) reported in numerous streams in the
northern hemisphere (Monteith et al., 2007; De Wit et al., 2021).
Numerous research carried out in temperate and boreal regions have shown that headwater
catchments are the main entry point of DOM into fluvial networks (Ågren et al., 2007; Creed

et al., 2015) and identified riparian areas as the dominant sources of DOM at the catchment scale owing to their location at the terrestrial-aquatic interface (Sanderman et al., 2009; Lambert et al., 2014; Laudon et al., 2012; Winterdahl et al., 2014). The flushing of shallow organic-rich soil layers during storm events (at the daily scale) typically represents the majority of annual DOC loads (Inamdar et al., 2006), and the DOC *versus* discharge relationships show that DOC export is transport-limited at the event scale (Buffam et al., 2001; Zarnetske et al., 2018). Although geomorphological and climatic conditions regulate DOC loads in aquatic ecosystems (Winterdahl et al., 2014; Laudon et al., 2012), DOC export at the annual scale is commonly conceptualized as a two-steps process in which DOM is produced and stored in the catchment during the hot and dry period, and then exported toward surface waters during the wet and cold period (Boyer et al., 1996). This two-steps conceptual model often described in temperate catchments (Deirmendjian et al., 2018; Strohmenger et al., 2020; Wen et al., 2020; Ruckhaus et al., 2023) is also supported by numerous studies carried out in tropical (Bouillon et al., 2014), boreal (Tiwari et al., 2022), Mediterranean (Butturini and Sabater, 2000) or Arctic fluvial networks (Neff et al., 2006). However the processes regulating the size of the riparian DOM pool remain unclear (Tank et al., 2018 and references below).

Antecedent soil conditions of wetness and temperature have been identified as a dominant control on stream DOC with concentrations typically increasing after dry events (Turgeon and Courchesne, 2008; Vázquez et al., 2007; Mehring et al., 2013). Periods of drought promote the production and accumulation of DOM in shallow soil horizons through enhanced soil organic matter decomposition (Harrison et al., 2008; Fenner and Freeman, 2011; Xu and Saiers, 2010), resulting in high stream DOC concentrations during the subsequent rewetting phase of the catchment (Werner et al., 2019; Raymond and Saiers, 2010). In good agreement with this conceptual model is the observation based on long-term data that the mean annual DOC concentrations in streams can be related to the intensity and duration of preceding dry periods (Humbert et al., 2015; Tiwari et al., 2022).

However, the establishment of reducing conditions in riparian soils during the winter may have potential implications on our conceptualization of stream DOC export owing to the influence of redox conditions on the iron (Fe) cycle in soils. While particulate Fe-hydroxides absorb organic substances with a high affinity when oxidizing conditions prevail, the microbially-driven dissolution of Fe oxyhydroxides during reducing conditions leads to the release of organic molecules previously bounded to surface minerals (Hagedorn et al., 2000; Blodau et al., 2008). The release of large amounts of DOM in riparian soils during the winter period – considered as non-productive in our current conceptualisation of stream DOC export – has been previously reported (Lambert et al., 2013; Lotfi-Kalahroodi et al., 2021), and

several studies have suggested that iron redox cycles may play a major role in catchment-scale DOC export (Knorr, 2013; Selle et al., 2019; Musolff et al., 2017). However, the onset of Fe reducing conditions and the subsequent DOM release could be limited in agricultural catchments owing to large inputs of nitrate (an oxidizing specie) from upslope via groundwater that may prevent Fe reductive biodissolution (Mcmahon and Chapelle, 2008; Christensen et al., 2000).

Because most of the studies investigating DOC export in headwater catchments rely on stream water monitoring, the processes regulating the size of the mobile DOM pool in riparian soils and the interaction with other biogeochemical cycles remain largely unknown. We still lack studies investigating how processes occurring in soil waters reflect our conceptualization of solutes dynamics based on observations made in surface waters (Knorr, 2013; Dupas et al., 2015; Ledesma et al., 2015; Seibert et al., 2009; Sanderman et al., 2009; Lambert et al., 2013). In this study, we hypothesized that Fe biodissolution may significantly affect DOM release in riparian soils during the winter period with consequences on stream DOC export. We also investigated the potential influence of nitrate from agricultural origin, which may regulate Fe reduction. To this end, we installed zero-tension lysimeters in the riparian area of the Kervidy-Naizin catchment, whose stream waters are continuously monitored for water quality, including DOC at high frequency (Fovet et al., 2018). This catchment is located in Brittany (France), a region where stream DOC concentrations exhibited contrasting trends (increasing, decreasing or no trend) over the 2007-2020 period despite similar geomorphological and climatic conditions (Supplementary Fig. S1). The Kervidy-Naizin catchment for instance exhibits a weak but significant increase in stream DOC concentrations over the last two decades (Strohmenger et al., 2020). In this context, another goal of this study was to explore the hypothesis that long-term regional decrease in nitrate inputs (Abbott et al., 2018) have impacted long-term trends in DOC through iron dynamics in riparian soils. We monitored soil water chemistry during the 2022-2023 hydrological year through measurements of DOC, Fe(II) and $NO_3$ concentrations but also DOM composition (absorbance and fluorescence properties coupled with parallel factor analysis) and soluble reactive phosphorus (SRP) as an additional tracer of Fe reductive dissolution (Gu et al., 2017; Smith et al., 2021). The results allowed us to decipher complex interactions among C, N, and Fe cycles in agricultural catchments and to highlight the occurrence of several processes sustaining DOM export during the winter period.

## 2. Material and method

### 2.1. Study site

The Kervidy-Naizin research observatory is a 4.9-km$^2$ agricultural headwater catchment located in Brittany (western France, Fig. 1). It belongs to the French Critical Zone Observatories (OZCAR) network and is instrumented since the 1970s for the long-term monitoring of the soil-atmosphere-hydrosphere continuum in a context of intensive agriculture (see Fovet et al., 2018 for a complete presentation of the study site).

The site is characterized by gentles slopes (<5%) and low elevation that ranges from 98–140 m above sea level. The bedrock is composed of impermeable Brioverian schists above which a locally fractured layer of schists is underlain by 1 – 30 m of weathered material and silty loam soils. Soils are well drained except in riparian zones, where water excess leads to hydromorphic, poorly drained soil. Soil organic carbon content presents lateral (riparian *versus* upland soils) and vertical (surface *versus* deep soils) gradients, with highest values about 5.3 – 5.6 % in the uppermost soil horizons (0-20 cm depth) of the riparian area while soil organic content drop under 1% below 20 cm depth (Lambert et al., 2011).

The land use is intensive mixed farming, with 91% of the catchment area under agriculture that grows crops to feed a high density of dairy cattle, pigs and poultry. Maize (38%), straw cereals (30%), and grasslands (15%) dominate and wooded areas are mainly confined to valley bottoms along the stream channel or to some hedgerows (Fig. 1).

The climate is temperate oceanic, with mean annual temperature of 11.2 ± 0.6°C and mean annual precipitation of 810 ± 180 mm. Precipitation varies seasonally throughout the year, with higher precipitation from October to February (mean monthly precipitation of 92 ± 31 mm) and lower precipitation from March to July (mean monthly precipitation of 50 ± 14 mm). The dynamics of the intermittent stream reflects the seasonal pattern of rainfall and evapotranspiration with high discharge periods from November to April and completely dry periods lasting one to three months between July to October depending on the hydrological year.

Groundwater level fluctuations are recorded every 15 min along the Kerolland (K) transect, rainfall is monitored at hourly intervals using a weather station located ~ 1400 m from the catchment outlet, and stream discharge is recorded every minute with an automatic gauge station at the outlet of the catchment. A S::SCAN probe is installed at the outlet of the catchment for the measurement of DOC and other variables at high-frequency (Fovet et al., 2018).

## 2.2. Monitoring and manual sampling

We investigated the seasonal variability in riparian DOM concentration and composition using zero-tension lysimeters designed to collect free soil waters (Supplementary Fig. S2) and installed in September 2022 in topsoil horizons (15 cm depth) in the Kerroland riparian

zone, an area known to be a major contributor to stream DOC export in this catchment
(Lambert et al., 2014). We placed the lysimeters along three lines parallel to the stream
channel, about 10-20 m apart from each other and from the stream, with the aim to capture
the heterogeneity of water flow paths and nitrate concentration coming from the upslope
cultivated fields. Lysimeters were all located in the hydromorphic soils unit according to the
soil map (Fig. 1). We installed 29 zero-tension lysimeters, but some were lost during the
study period because of damage by rodents. We kept lysimeters for which at least seven
consecutive dates were available, resulting in 17 lysimeters used for the study. We collected
soil waters from November 2022 to June 2023 at a weekly to fortnightly frequency depending
on the hydro-climatic conditions (Fig. 2). The end of sampling was imposed by the lack of
water in lysimeters owing to the gradual drawdown of the water table in the riparian zone
during the spring period. We sampled soil waters with a vacuum pump and filtered them at
0.2 µm with acetate cellulose syringe encapsulated filters directly on site for all analyses
including DOC, $NO_3$, SRP, Fe(II), and DOM composition (absorbance and fluorescence). We
used unfiltered water samples to measure physico-chemistry variables including temperature
and pH with an ODEON probe. In addition, we collected surface waters right next to the
riparian area where lysimeters were located and at the outlet of the catchment. The
laboratory analyses were identical for soil and surface waters.

### 2.3. Analytical procedures

With the exception of Fe(II) measurements that were performed the same day as sampling,
all analyses were done within two weeks after sampling. Samples were stored in a 4°C cold
room in the dark. Fe(II) analyses were determined using the 1.10-phenanthroline colorimetric
method (Lambert et al., 2013): dissolved iron was trapped on site and the optical density of
the complex formed with phenanthroline was measured the same day once back to the
laboratory at 510 nm with an UV-vis spectrophotometer. DOC concentrations were measured
using a total carbon analyzer (SHIMADZU TOC-V) with a precision estimated at ± 5% using
a standard potassium hydrogen phthalate solution (SIGMA ALDRICH). Nitrate as N-$NO_3^-$ and
SRP were determined by spectrometry with an automatic sequential analyzer (SmartChem
200, AMS Alliance, France).
Absorbance for colored DOM (CDOM) was measured with a Lambda 365 UV/vis
spectrophotometer (Perkin Elmer) from 200 to 700 nm (1 nm increment) using a 1 cm quartz
cuvette. Samples were diluted in most case due the DOM-rich nature of soil waters. The only
purpose of CDOM spectra was to correct excitation-emission matrices (EEMs) for inner filter
effects (Ohno, 2002). The dilution factor used for fluorescence measurements were applied
to CDOM spectra. Fluorescence DOM (FDOM) was collected as EEMs with a Lambda LS45
(Perkin Elmer) using a 1 cm quartz cuvette across excitation wavelengths of 270 – 450 nm (5
nm increment) and emission wavelengths of 290 – 600 nm (0.5 nm increment). Samples
were diluted so absorbance at 254 nm was below 0.3 to reduce inner filter effects (Ohno,

177 2002).

In our study, the Fe(II):DOC ratio was 0.30±0.24, implying that significant interferences on
DOM fluorescence from iron can be expected (Poulin et al., 2014). The degree of iron
quenching, however, varies greatly between samples depending on the iron:DOC ratio
(Pullin et al., 2007) but also on DOM composition (Jia et al., 2021; Poulin et al., 2014) and
Fe(III) concentrations (Ohno et al., 2008), making difficult to predict the influence of Fe on
EEMs. That being said, quenching was clearly apparent in some samples (n < 10) that
showed the fluorescence intensity to increase with dilution factor, reflecting the influence of
high level of Fe that reduces DOM fluorescence (Pullin et al., 2007). The quenching
impacted EEMs at low (< 270 nm) and moderate to high (420 – 490 nm) excitation and
emission wavelengths, respectively, which is consistent with previous studies concluding that
Fe mainly impacts fluorescence intensity in EEM locations associated with humic-like
fluorophores, namely A and C peaks (Jia et al., 2021; Poulin et al., 2014). Thus, although we
cannot rule out an effect of iron on EEMs, this would have impacted the relative contribution
of humic-like fluorophores associated with C1 and C2 components of our model (see below)
who behaved similarly between clusters and across seasons.

### 193    2.4. PARAFAC modelling

EEMs preprocessing (Raman scattering removal and standardization to Raman units) was
performed prior to the PARAFAC modeling. Normalization was done using a Milli-Q water
sample run the same day as the sample. A five-component PARAFAC model was obtained
using the drEEM 0.3.0 Toolbox (Murphy et al., 2013) for MATLAB (MathWorks, Natick, MA,
USA). Split-half analysis, random initialization, and visualization of residuals EEMs were
used to test and validate the model. The positions of maximum peaks of the PARAFAC
components were compared to previous studies carried out in similar context of human-
impacted catchments with the open fluorescence database OpenFluor using the OpenFluor
add-on for the open-source chromatography software OpenChrom (Murphy et al., 2014). The
maximum fluorescence $F_{Max}$ values of each component for a particular sample provided by
the model were summed to calculate the total fluorescence signal $F_{Tot}$ of the sample in
Raman units. The relative abundance of any particular PARAFAC component X was then
calculated as $\%C_X = F_{Max}(X)/F_{Tot}$.

### 207    2.5 Statistical Analyses

A principal component analysis (PCA) coupled to a clustering analysis was used to discriminate and group lysimeters based on the presence or absence of iron biodissolution in soil waters. The aim was to help visualize temporal pattern for each of the two clusters rather than 17 time series if data were plotted for each lysimeter. For this reason, data (DOC, $NO_3$, SRP and Fe(II) concentrations and the relative contribution of PARAFAC components) were averaged for each lysimeters then normalized. The PCA was performed using the *prcomp* function in the R software, and the *factoextra* package was used to identify the variables that contribute the most to the first two dimensions of the PCA. The cluster analysis, based on the results from the PCA and called Hierarchical Clustering on Principal Components (Josse, 2010), was performed with the *FactoMineR* package for R (Lê et al., 2008). Relationships between variables were investigated either through Pearson or Spearman correlations depending of the nature (linear or not) of the correlations.

## 3. Results

### 3.1. Hydro-climatic context

The hydrological regime of the study site is characterized by a succession of three distinct periods determined by water table fluctuations along the hillslope, corresponding to different hydrological regimes for the riparian soils (Fig. 2; Lambert et al., 2013): (i) a period of progressive rewetting of riparian soils after the dry season and of low groundwater flow and low stream discharge (01/09/2022 – 18/12/2022, mean and cumulated precipitation = 5.1±5.3 mm $d^{-1}$ and 338.5 mm, respectively); (ii) a period of prolonged waterlogging of riparian soils induced by the rise of the water table in the upland domain, corresponding to high values of hillslope groundwater flow and stream discharge (18/12/2022 – 9/05/2023, mean and cumulated precipitation = 6.8±7.9 mm $d^{-1}$ and 573 mm, respectively); and (iii) a period of drainage and progressive drying of the riparian soils induced by the drawdown first in the upland domain then in the bottomland domain and corresponding to the decrease of both the hillslope groundwater flow and stream discharge (09/05/2023 – 01/07/2023, mean and cumulated precipitation = 4.3±4.4 mm $d^{-1}$ and 42.5 mm, respectively). Air temperature (Fig. 2C) showed a smoothed seasonal variability with decreasing values from September to December (from ~20°C to -2°C) followed by a rise in temperature from 0°C to 20°C from February to July. This pattern was only interrupted by a relatively short episode of higher temperature (close to 10°C) during the winter, coinciding with the first intense rainfall period of the year.

### 3.2. Fluorescence properties of DOM

Five PARAFAC components were identified in soil waters (Supplementary Fig. S3), all of which already described in previous studies. All five components had humic-like fluorescence

properties (Fellman et al., 2010). Components C1 (excitation/emission peaks = 350 nm /444 nm), C2 (<270/450), and C5 (410/488) predominantly cover the regions of EEMs associated with peaks A and C and are common tracers of terrestrially-derived DOM in surface waters (Kothawala et al., 2015; Stedmon and Markager, 2005; Logozzo et al., 2023; Lambert et al., 2017) while C3 (330/406) and C4 (295/410) are both located near the classical peak M, indicating a microbial transformation of terrestrial DOM (Williams et al., 2010; Lambert et al., 2022; Yamashita et al., 2010). The maximum fluorescence intensity of all components were strongly related to DOC concentrations (not shown) and the relative contribution of each component decreased from as C1 (29.7±3.1 %) > C2 (28.3±3.6 %) > C3 (19.5±2.5 %) > C4 (12.9±6.6 %) > C5 (9.7±2.1 %).

### 3.3. Seasonal variations in soil and stream waters

Temperature in soil waters (Fig. 3A) followed the same pattern as air temperature: values oscillated between 5°C and 15°C during November – January, reached minimums between 4 and 7 °C in January – March and then increased gradually during the end of the study period up to 18 – 20 °C in June. pH varied between 6.2 and 7.4 (mean 6.9 ± 0.3) across lysimeters and didn't exhibit significant trends over the study period (Fig. 3B). Solutes, however, exhibited complex patterns with a high variability across lysimeters and time, especially during the high flow period (Fig. 3C-F). Despite the fact that lysimeters were installed along three lines ranging 10-30 m from the stream, no spatial pattern was identified. Overall, these elements were strongly linked to each other (Fig. 4). DOC concentrations ranged from 2.3 to 87.4 mg $L^{-1}$ (mean = 30.2±12.8 mg $L^{-1}$) over the study period and were linearly and positively (Pearson r = 0.73, $p$ value < 0.0001) associated with Fe(II) that ranged from 0 to 45.8 mg $L^{-1}$ (mean = 9.8±7.6 mg $L^{-1}$). Fe(II) was negatively (Spearman r = -0.56, $p$ value < 0.0001) correlated with $NO_3$ (from 0 to 16.4 mg $L^{-1}$, mean = 0.9±1.1 mg $L^{-1}$), and SRP (from 0 to 0.5 mg $L^{-1}$, mean = 0.1±0.1 mg $L^{-1}$) was also positively (Pearson r = 0.21, $p$ value = 0.0005) related to Fe(II), but not as strongly as for DOC.

DOC concentrations in stream waters varied from 2.9 to 36.8 mg $L^{-1}$ during the study period (Fig. 5). Maximum concentrations were reached during storm events due to a rapid response to rainfall and the mobilisation of riparian wetland waters (Durand and Juan Torres, 1996). There was a tendency for minimum (at base flow) and maximum (at peak discharge) concentrations to decrease from November to March. From March to July, however, minimal concentrations remained stable while maximum values showed a slight increasing trend.

### 3.4. Clustering of soil waters

The first two components of the PCA explained 69.4 % of the total variance of the data and discriminated lysimeters depending on the presence or absence of Fe(II) biodissolution in

soil waters of the riparian area (Fig. 6). The first principal component (PC1, 54% of the total variance) was mainly related to $NO_3$ concentrations and terrestrial humic-like components (C1, C2, and C5) on positive scores, and to DOC and Fe(II) concentrations and the microbial humic-like component C4 on negative scores. The second component (PC2, 15.4% of the total variance) was related to SRP (positive score) and the component C3 (negative score). PARAFAC components had similar or even higher scores than DOC, Fe(II), and $NO_3$ concentrations on the two first dimensions of the PCA (Supplementary Fig. S4), illustrating the importance of DOM composition as an important factor contributing to explain the spatial variability across lysimeters. The distribution of PARAFAC components along the first dimension reflects the relationships between their relative contribution and Fe(II), concentrations (not shown). More specifically, %C4 was strongly and positively correlated with Fe(II) ($R^2 = 0.38$, Pearson r $= 0.62$) compared to other components that exhibited weakest and negative relationships with Fe(II) ($R^2$ from 0.09 to 0.19, Pearson r from -0.30 to -0.43). In other words, lysimeters capturing Fe biodissolution in the riparian area were associated with high DOC and a greater proportion of the microbial C4 component compared to lysimeters enriched in nitrate where no Fe(II) was measured.

The hierarchical clustering based on the PCA results grouped the lysimeters in two distinct clusters based on the presence (cluster 1) or absence (cluster 2) of Fe(II) (Fig. 6). This approach allowed us to gain insight into the temporal evolution of solutes in soil waters since clear patterns appeared once the data were grouped by cluster (Fig. 7). In cluster 1, DOC, N-$NO_3$ and SRP decreased from 39.8±13.3 to 23.4±8.4 mg $L^{-1}$, from 2.6±3.6 to 1.2±1.8 mg $L^{-1}$, and from 0.18±0.18 to 0.08±0.15 mg $L^{-1}$, respectively, during the rewetting phase of the catchment while Fe(II) was no measured at significant levels. During the high flow period, however, Fe(II) increased gradually from 3.7±3.2 to 26.5±7.8 mg $L^{-1}$, and both DOC and SRP followed a similar trend with concentrations raising from 27.3±9.5 to 54.9±25.0 mg $L^{-1}$ and from 0.07±0.13 to 0.18±0.11 mg $L^{-1}$, respectively. During this period and until the end of the hydrological cycle, N-$NO_3$ were very low, decreasing from 0.54±0.66 mg $L^{-1}$ at the beginning of the high flow period to values below 0.15 mg $L^{-1}$ the rest of the survey. The start of the third hydrological period corresponding to the drawdown of the water table and the consecutive aeration of riparian soils was marked by the rapid drop of Fe(II) at 8.1±7.4 mg $L^{-1}$, DOC at 17.5±10.9 mg $L^{-1}$, and SRP at 0.02±0.02 mg $L^{-1}$.

Similarly to cluster 1, soil waters from the cluster 2 exhibited a decline in DOC and SRP concentrations during the rewetting phase of the catchment but these trends continued during the high flow period, with minimal values reached in the middle of February. Thus, DOC dropped from 34.5±7.1 to 9.4±3.1 mg $L^{-1}$ and SRP from 0.19±0.08 to 0.02±0.01 mg $L^{-1}$ during this period, before showing an increasing trend to reach concentrations about

21.0±6.1 mg $L^{-1}$ for DOC and 0.16±0.13 mg $L^{-1}$ for SRP at the end of the high flow period.
DOC remained elevated (24.1±3.1 mg $L^{-1}$) at the start of the dry period, but SRP dropped
close to depletion. In contrast, $N-NO_3$ first increased from 0.57±0.81 mg $L^{-1}$ in November to
maximum values of 6.5±5.9 mg $L^{-1}$ in the middle of March, and then exhibited decreasing
concentrations until a complete depletion at the beginning of the third hydrological period.
Contrary to cluster 1, Fe(II) was not measured at significant concentrations in cluster 2 (*i.e.*
below 0.5 mg $L^{-1}$) except in March, during which Fe(II) increased from 1.2±1.9 to 4.1±0.2 mg
$L^{-1}$.
**4. Discussion**
**4.1. The buffering effect of nitrate on iron reductive dissolution**
The reductive biodissolution of iron during the high-water winter period is a recurrent process
in riparian soils of headwater catchments (Smolders et al., 2017; Knorr, 2013; Selle et al.,
2019). The magnitude of variations in Fe(II) and associated DOC and SRP dynamics
reported in this study are in line with previous works conducted in the same research
catchment (Lambert et al., 2013; Lotfi-Kalahroodi et al., 2021; Gu et al., 2017). In addition,
our results evidenced a marked variability in the intensity of iron dissolution across lysimeters
that we attributed to the spatial distribution of $NO_3$-rich water flow paths that can inhibit and
delay the release of Fe(II) and DOC in soil waters.
A fundamental condition for the establishment of reductive conditions is the prolonged
waterlogging of riparian soils. As shown earlier for this and other lowland catchments on
impervious bedrock, the increase of the hydraulic gradient induced by the rise of
groundwater in the upland domain during the high flow period maintains a strong hydrologic
connection between upland and riparian domains (Pacific et al., 2010; Molenat et al., 2008).
Under these conditions, riparian soils remain waterlogged owing to a high and continuous
hillslope groundwater flow, leading to the gradual establishment of reductive conditions and
the subsequent triggering of Fe-biodissolution as long as inputs of oxidizing species
remained limited and/or counterbalanced by higher rate of consumption through microbial
activity (Lotfi-Kalahroodi et al., 2021; Lambert et al., 2013). This pattern is well illustrated by
data from lysimeters of the first cluster (Fig. 7). After a quick depletion of an initial stock of
nitrate accumulated during the previous summer, reductive conditions were rapidly
established at the beginning of the high flow period and increasing Fe(II) concentrations in
soil waters lead to the onset of the reductive Fe biodissolution in riparian soils. The gradual
increase in Fe(II) during all the high flow period despite variations in temperature or rainfall
patterns (with some intense precipitation events > 20 mm $d^{-1}$) suggests a limited impact of
these climatic episodes, except during a period of low precipitation during which both Fe(II)
and DOC exhibited a slight decrease in February/March. We attributed this small drop to the
drawdown of the water table in upland groundwater flow following a prolonged absence of
precipitations (see PK3 fluctuations, Fig. 2) that may have re-oxygenated soil waters (as no
changes in N-$NO_3$ occurred).
Therefore, large release of DOC occurred in soils of the first cluster. Iron biodissolution also
affected SRP, but the relationships was weaker suggesting that the reductive dissolution of
soil Fe was not the primary driver of SRP concentrations in soils. For instance, soil
properties, and more specifically soil phosphorus content and speciation, have been shown
to strongly regulate SRP in soil waters of the Kervidy-Naizin catchment (Gu et al., 2017).
Regarding DOC, the mean DOC:Fe(II) molar ratio was 142.4±285.5. This was higher than
the DOC:Fe(II) ratio measured in experimental conditions (74.5±74.6) but similar to value
measured on the field (134.4±25.6) by Lotfi-Kalahroodi et al. (2021) who aimed to investigate
Fe reduction in the riparian area of our study catchment. Fe(III) concentrations in soil waters
were not measured, but, based on the work of Lotfi-Kalahroodi et al. (2021), we can estimate
a ratio between total Fe and Fe(II) of 4.8. Keeping in mind that this is a rough estimation, our
mean DOC:Fe ratio would be about 29.3±58.8, which is consistent with previous studies (e.g.
Selle et al., 2019; Musolff et al., 2017; Grybos et al., 2009; Cabezas et al., 2013). The nature
of processes releasing DOC upon the reduction of soil-Fe oxyhydroxides in riparian soils of
our study site has been studied in laboratory conditions (Grybos et al., 2009). Results have
shown that up to 60% of the release is due to DOC desorption caused by the pH increase
that accompanies the reduction of Feoxyhydroxides in these soils, the remaining 40% being
due to the dissolution of Fe-oxyhydroxides that strongly adsorb organic compounds
previously bounded to surface minerals (e.g. Hagedorn et al., 2000). In good agreement with
these results, soil DOC was positively related to pH (Supplementary Fig. S5). The abrupt
decrease in DOC in June illustrates the restoration of aerobic conditions owing to the
drawdown of the water table in the bottomland domain led to the formation of Fe-minerals
and the subsequent retention of DOC and SRP (Gu et al., 2017).
Lysimeters from the second cluster showed a very different pattern. Although some of them
were located close (3-4 m) to lysimeters in which reducing conditions prevailed, there was no
evidence of Fe(II) release, arguably because of the presence of nitrate. Indeed, and in
agreement with studies carried out in wetland (Lucassen et al., 2004) and lacustrine
(Andersen, 1982) sediments, we argue that the Fe-biodissolution biodissolution was inhibited
as long as long as $NO_3$ remained in sufficient quantity in soil waters. In the absence of such
production or regeneration process, both DOC and SRP showed a net depletion pattern from
November to March. The influence of nitrate as a buffer of Fe-biodissolution was furthermore
supported by the observation of a slight release of Fe(II) in May, at a moment when nitrate

became depleted from soil waters, probably because of plant uptake. Interestingly, we found that the threshold value of nitrate above which the process is activated (based on the $NO_3$ versus Fe(II) relationship (Fig. 4) as well as timing of Fe-biodissolution identified in cluster 1 and cluster 2) ranged between 1.2 and 1.8 $N-NO_3$ (4.1 – 6.2 mg $L^{-1}$), which is close to the threshold value of 6 mg $L^{-1}$ established at the catchment scale by Musolff et al. (2017) in German streams.

The PARAFAC components identified in the model suggest a dominance of highly aromatic and conjugated molecules across all lysimeters and dates, which is typical of DOM derived from soil organic matter and found in poorly drained soils in riparian or wetland areas (Sanderman et al., 2009; Lambert et al., 2013; Yamashita et al., 2010). The larger proportion of C4 in the first cluster however indicates that the Fe oxyhydroxides reduction leads to greater proportion of microbially-derived compounds within the DOM pool. In agreement with previous studies showing that the Fe(III) reduction could enhance the decomposition of organic matter in soils (Chen et al., 2020; Kappler et al., 2021), the close link between Fe(II) and C4 likely reflects an indirect effect of Fe biodissolution promoting the degradation of soil OM and the subsequent incorporation of microbially-derived compounds into the DOM pool (Dong et al., 2023). This hypothesis is well consistent with previous experimental studies performed with soils from the Kervidy-Naizin riparian area, which showed that bacterial reduction of Fe(III)-oxides to Fe(II) was concomitant with the release of large biological organic by-products upon the growth of bacterial communities (Lotfi-Kalahroodi et al., 2021).

Our study evidences a strong spatial heterogeneity of the establishment of reducing conditions in the riparian area of the Kervidy-Naizin catchment, associated with differences in the composition of DOM released in soil waters. It remain to be determined, however, the reason for such variability in biogeochemical processes in riparian soils. A first explanation can be related to the heterogeneity in water flowpaths in soils. In intensive agricultural catchments such as our study site, inflow of $NO_3$-rich water may arise from the rise of contaminated groundwater in valley bottom s and/or from subsurface flow paths that connect upland soils to riparian soils (Molenat et al., 2008). It is likely that lysimeters from the second cluster captured preferential flow paths of $NO_3$-rich waters while lysimeters from the first cluster were disconnected from those preferential water circulations. Alternatively, the absence of nitrate in soil waters may arise from a higher rate of denitrification that counterbalanced $NO_3$ inputs. Research based on field observation remained limited to decipher the respective role of hydrology versus biogeochemistry in controlling Fe(II) biodissolution in riparian soils, and experimental studies would be required to provide more quantitative values on these potential drivers and their interactions.

## 4.2. Implication for stream DOM export at the catchment scale

The current understanding of DOM export in headwater catchments is based on a two-steps conceptual model, in which a pool of mobile DOM is built in soils during the dry season and then flushed towards surface waters during the following wet season (e.g. Tiwari et al., 2022; Ruckhaus et al., 2023; Strohmenger et al., 2020; Raymond and Saiers, 2010). However, the high-frequency measurements of DOC in the stream do not fully support this statement. The establishment of a hydrological connection between riparian soils and the stream during the winter period showed the stream DOC to gradually decrease both at peak discharge during successive storm events and at base flow during inter-storm periods (Figure 5). This pattern, which repeats every year in this catchment (Strohmenger et al., 2020), is well consistent with the hypothesis of the mobilisation and exhaustion of a DOM pool limited in size built during the summer period (Humbert et al., 2015). However, stream DOC were found to increase slightly in March/April after the low-flow period that showed the hydrological connection between soils and the stream to decrease. It is unlikely that the mobilisation of an additional pool of DOM from upland soils may explain this small raises in stream DOC because this pool is 1) relatively small in terms of size, and 2) quickly exhausted at the beginning of the winter period (Lambert et al., 2014). Therefore, the seasonal pattern of stream DOC likely reflects the regeneration of the riparian DOM pool during the winter period as shown by our data collected in soil waters of riparian wetlands.

Stable carbon isotopes have indeed demonstrated that riparian soils of the Kervidy-Naizin catchment – and more particularly the DOM-rich uppermost soil horizons – are the dominant source of stream DOC at the catchment scale (Lambert et al., 2014), a feature commonly shared by headwater catchments (e.g. Sanderman et al., 2009). Thus, the decline in DOC and SRP observed in soil waters, particularly in the second cluster whereby these elements became almost depleted (Fig. 7), was consistent with the general flushing behaviour of the catchment shown by stream DOC from November to February. Similarly, the large two to three fold increase in DOC concentrations in riparian soils (in cluster 1 and 2, respectively) denotes a large mobilisation of DOM between March and May despite wet and low temperature conditions, that could explain in turn the pattern observed in stream DOC at the same time. While part of this regeneration can be attributed to iron biodissolution, the release of large amount of DOC the cluster 2 where the reductive biodissolution of Fe(III) was limited implies that another production mechanisms contributed to release DOM in riparian soils. It is unlikely that agricultural inputs (crop residues, manure application, etc) main may explain the increases in the riparian area, as these sources are episodic and/or size-limited (Lambert et al., 2014; Humbert et al., 2015; Pacific et al., 2010). This observation echoes previous works on the Kervidy-Naizin catchment showing effective inter-annual regeneration mechanisms of

the pool of soluble phosphorus in soils unrelated to iron dynamics (Gu et al., 2017), a
statement supported here by the fact that SRP concentrations followed a similar pattern as
DOC in soils grouped in the second cluster (Fig. 7).
The PARAFAC results suggest that DOM mobilized from soil to streams is only composed by
aromatic molecules of high molecular weight. Although complex organic molecules indeed
dominate stream DOM export (Fellman et al., 2009), it should be noted however that protein-
like components are commonly found in stream waters (Inamdar et al., 2012), including in
our study site (Humbert et al., 2020). The lack of such components in our model results from
our sampling approach and not from their absence in catchment soils. Indeed, the production
of protein-like components in catchment soils is restricted to the summer hot and dry period
during which a pool made of low-aromatic and microbially-derived compounds built up in
riparian soils (Lambert et al., 2013). However, this DOM pool is quickly flushed and
exhausted during the rewetting phase in October-November, and soil DOM during the winter
period is mainly composed by highly-aromatic molecules originating from soil organic
material (Lambert et al., 2014). Agricultural practices such as fertilizer applications can
represent another source of protein-like DOM in the catchment (Humbert et al., 2020), but
these inputs remain episodic with a low impact on DOM at the catchment scale (Humbert et
al., 2015; Lambert et al., 2014). For instance, a recent one-year of monitoring of soil waters
at different locations in the catchment has shown that protein-like components represent only
$3.44 \pm 2.8\%$ of the total fluorescence signal in catchment soils, this contribution being
particularly low in riparian areas (Humbert et al., 2020). Therefore, the absence of protein-
like components in our PARAFAC model is the consequence of our sampling design that
focused on DOM production mechanisms in riparian soils (distant from agricultural inputs)
during the winter period (period of production of highly aromatic compounds in soils).
Taking together, our results have two important implications regarding our conceptualisation
of DOM export in headwater catchments. First, it challenges the idea that the wet period acts
solely as a passive export period for DOC, with no or little DOC production (Strohmenger et
al., 2020; Ruckhaus et al., 2023; Wen et al., 2020). Second, it emphasis that stream DOC
dynamics at the outlet is an integrative signal, potentially masking the high spatial
heterogeneity of the system owing to complex interactions between biogeochemical cycles in
soils, nutrient transfer at the soil/stream interface and hydrological functioning of catchments.
While the patterns of stream DOC were consistent with that observed in soils, our study
remains however limited in its capacity to quantify the relative contribution of the cluster
identified to stream DOC export. Additionally, we do not have the necessary data such as
isotopes or molecular markers to elucidate the precise origin and DOM (and SRP) release in
soils unrelated to iron biodissolution, and this should be the focus of future work combining
experimental and field studies.

**Conclusion**

The combined monitoring of soil and stream waters in a temperate headwater catchment
allowed us to evidence the dual role of high flow period as both an active phase of DOC
production and export. In agreement with previous studies (e.g. Selle et al., 2019; Knorr,
2013), the establishment of Fe-reducing conditions in riparian areas was identified as a major
mechanism for the release of large amount of DOM in soil waters. In agricultural catchments,
however, we found that this process can be buffered by nitrate, leading to a strong spatial
heterogeneity in the magnitude of iron biodissolution and its consequences on soil DOC
dynamics. Our study also evidenced that another production mechanisms unrelated to Fe
dynamics contributed to release DOM in riparian soils during the winter period, pointing to
the need to further investigate stream DOC export at the soil/stream interface.
The interactions between the N and Fe biogeochemical cycles may have potential
implications regarding long-term increases in DOC in streams of Brittany. Indeed, stream
DOC in the Kervidy-Naizin catchment has been slowly but significantly increasing in the last
two decades, and this trend is mirrored by a decline in $NO_3$ concentrations (Strohmenger et
al., 2020). While part of the DOC trend can be related to changes in climatic conditions as
winters tend to wetter over the years (Strohmenger et al., 2020), the long-term decline in N
inputs from agriculture may have favoured the increase in stream DOC by enhancing Fe(II)
biodissolution in riparian soils. This hypothesis could partly explain why catchments having
similar geomorphological and climatic properties present contrasting long-term trends at the
scale of the Brittany region (Supplementary Fig. S1). Indeed, nitrate concentrations have
largely decreased during the last decades, but the rate of recovery is not uniform across the
region (Abbott et al., 2018). Studies carried out at the regional scale aiming to decipher the
interactions between local (agricultural practices) and global (climatic conditions) and the
consequences on stream DOC export would be critical considering the influence of DOM on
water quality and on the ecological and biogeochemical functioning of surface waters.

**Data availability**

Data on soil waters will be published on Zenodo.org upon the reservation that the paper will
be accepted for publication. Hydrological and climatic data from the Kervidy-Naizin site are
available here: https://geosas.fr/web/?page_id=103.

**Acknowledgements**

We thank Militza G., Harald F., P. Petitjean and Celine B. for their assistance in field and lab
work.

**Financial support**

This study has received funding from the H2020 European Research Council under the Marie
Skłodowska-Curie grant agreement COSTREAM No 101064945.

**Author contribution**

TL conceived the study. TL defined protocols with contribution from RD and PD. TL collected
field samples with help from RD. TL made laboratory analysis. TL analysed the data and
drafted the manuscript with inputs from RD and PD. All authors contributed and approved to
the manuscript.

**Competing interests**

The authors declare that they have no conflict of interest.

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

**Figure Caption**
**Figure 1 –** Location map of the Kervidy-Naizin experimental catchment showing land uses.
Hatched areas located along the stream channel network indicate the extent of hydromorphic
soils commonly waterlogged during the winter period. Lysimeters were located downslope the
piezometer PK1.
**Figure 2 –** (A) Record of hourly discharge and daily rainfall, (B) record of hourly piezometric
levels in wetland (PK1) and upland (PK3) domains, and (C) record of daily air temperature.
Black triangles in panel A indicate fieldwork for manual sampling of soil and stream waters.
Vertical black dashed lines delimit the different hydrologic periods, namely the rewetting, high
flow, and recession phases. See text for details.
**Figure 3 –** Evolution of (A) air temperature and (B) pH, (C) DOC, (D) $NO_3$, (E) Fe(II), and (F)
SRP in soil waters during the study period. Vertical black dashed lines delimit the different
hydrologic periods, namely the rewetting, high flow, and recession phases. See text for details.
**Figure 4 –** Relationships between (A) DOC and Fe(II), (B) Fe(II) and $NO_3$, ad (C) SRP and
Fe(II) in soil waters during the study period.
**Figure 5 –** Variations in stream DOC measured at high frequency at the outlet of the
catchment. Vertical black dashed lines delimit the different hydrologic periods, namely the
rewetting, high flow, and recession phases. See text for details.
**Figure 6 –** PCA biplot, including loadings plot for the input variables and scores plot for
lysimeters. One point represents one lysimeters, PCA being based on average values
calculated over the study period. Markers are coloured according to the cluster identified by
the Hierarchical Clustering on Principal Components (see material and methods).
**Figure 7 –** Evolution of (A) DOC, (B) Fe(II), (C) $NO_3$, and (D) SRP in soil waters for each
cluster. Lysimeters are grouped according the Hierarchical Clustering on Principal
Components (see text for details and Fig. 6). Vertical black dashed lines delimit the different
hydrologic periods, namely the rewetting, high flow, and recession phases. See text for details.


**Figure 1**

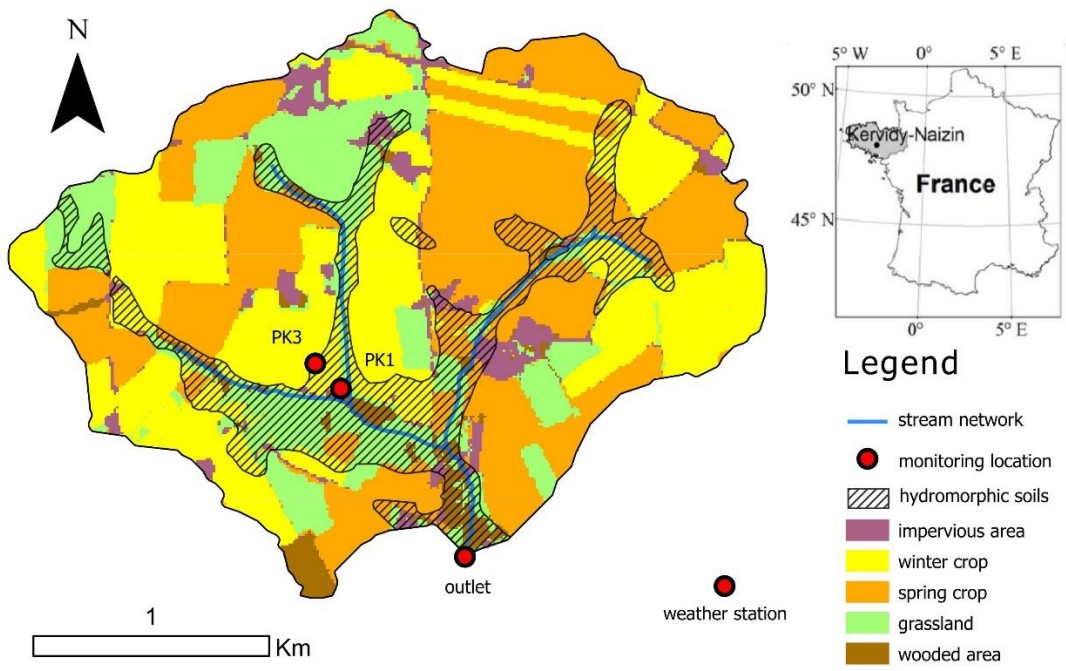





**Figure 2**

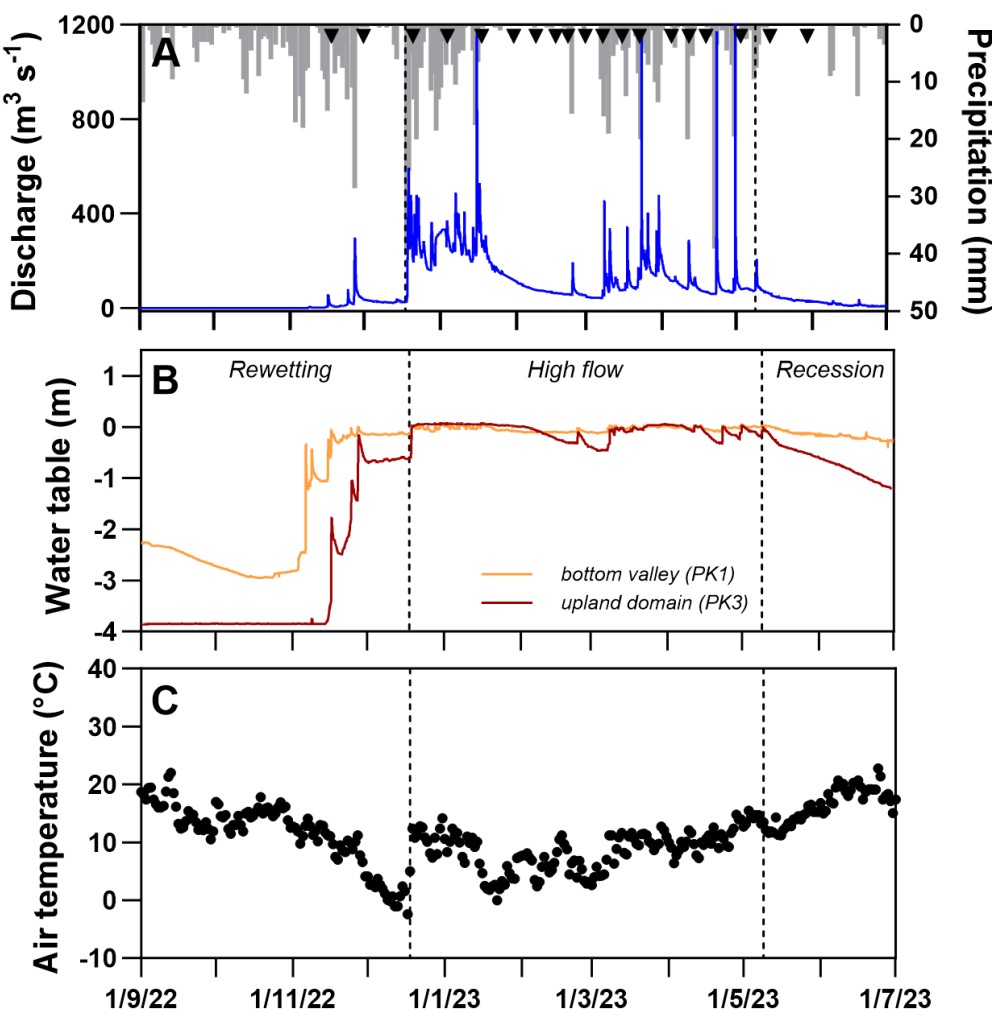

**Figure 3**

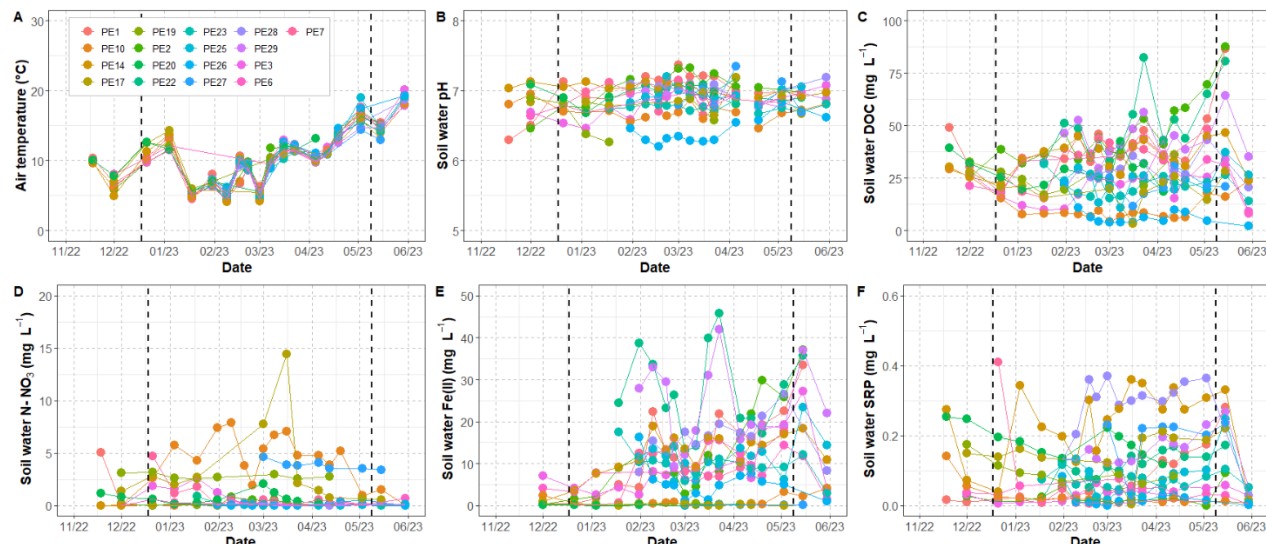



**Figure 4**

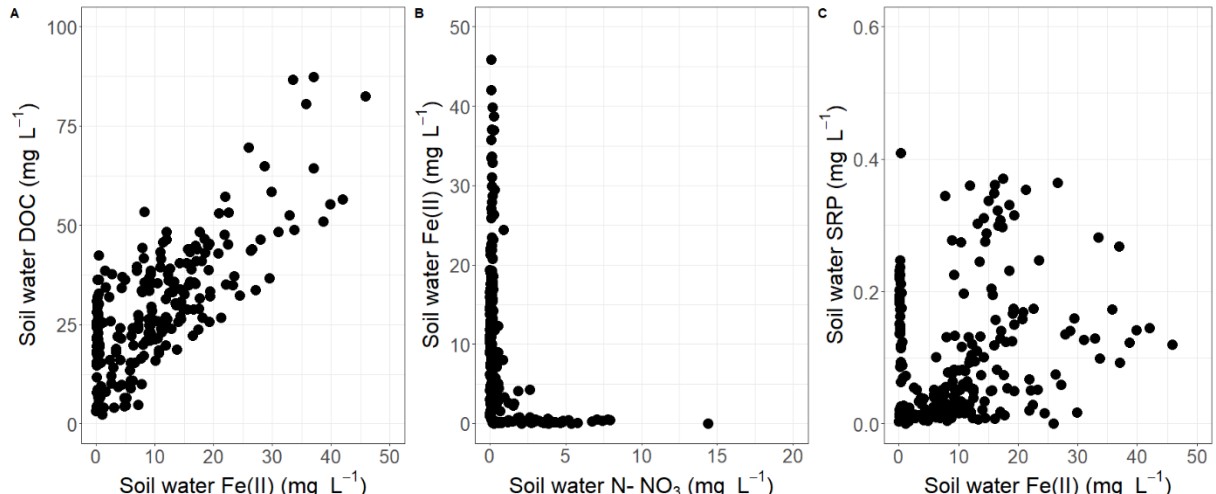



**Figure 5**

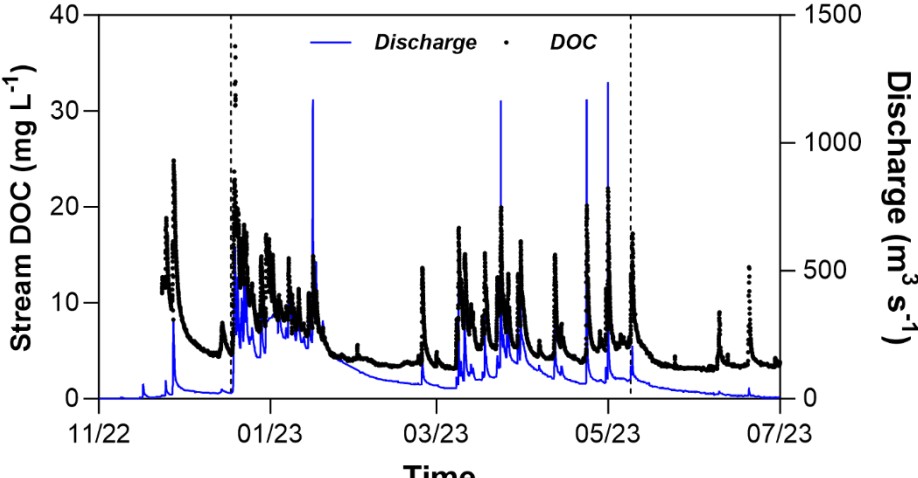




**Figure 6**

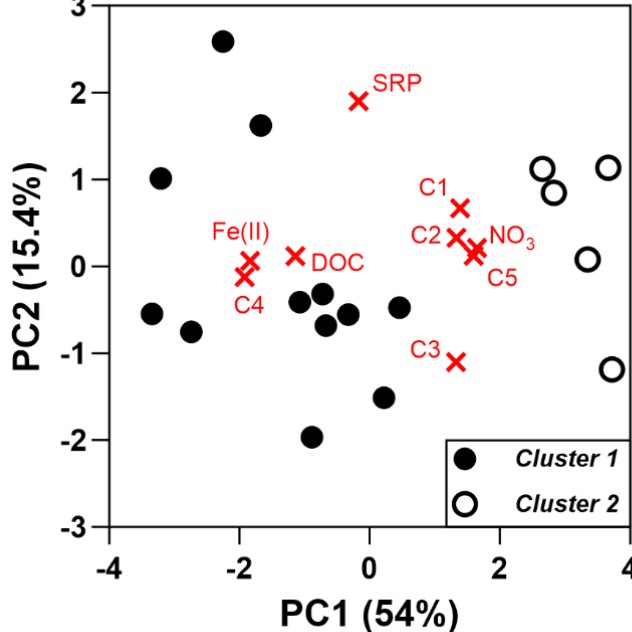




 **Figure 7**

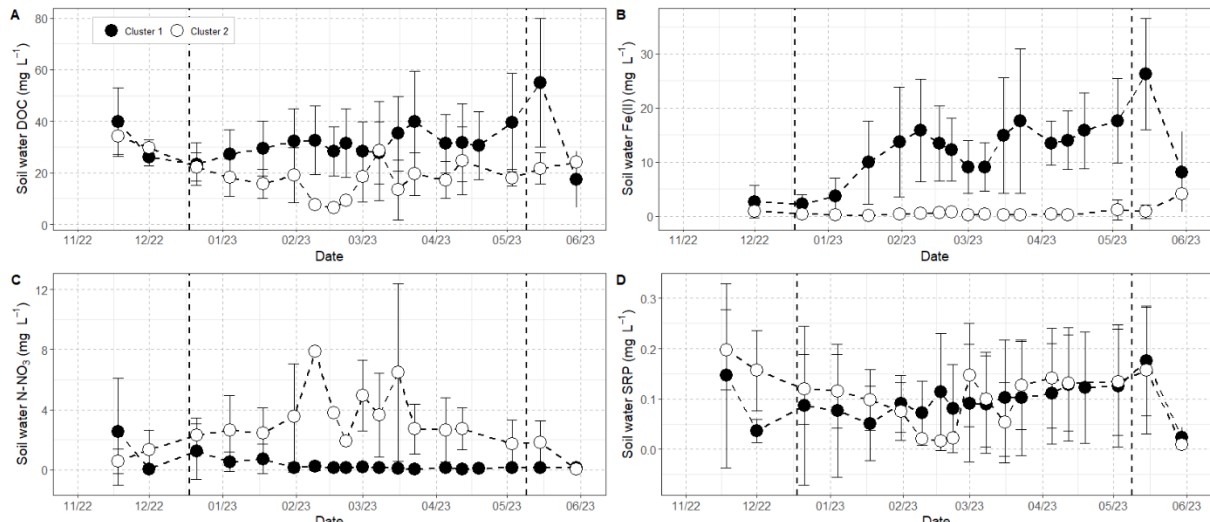