# Peer review of "The role of nitrogen and iron biogeochemical cycles on the production and"

_EGUsphere, 2024_

## Author Comment (AC2)

**General comments**

This manuscript presents results from a sampling campaign in the riparian zone shallow groundwater and draws conclusions on the redox conditions influencing DOM exports into the stream of a small agricultural catchment. This work builds on preceding works in the same catchment and extents the previous findings and hypotheses. The manuscript is a good match for HESS and should be of interest for scientists working on catchment water quality.

The manuscript is written in a concise way. Figures are mostly good and references up to date. My specific comments given below add up to a quite long list but are not substantial. My most critical point is the temporal averaging of lysimeter data that nees a better justification. However, from my point of view some work is needed to bring this manuscript into a final acceptable form.

**REPLY:** We thank the reviewer for their positive evaluation of our work.

**Specific comments**

*Abstract*

The abstract uses DOC while title and manuscript text uses DOM. Homogenize that?

**REPLY:** The abstract was modified as followed:

" […] we combined monitoring of nitrates, iron, soluble phosphorus and dissolved organic matter concentration (as dissolved organic carbon; DOC) […] "

L7: Check this first sentence. Not clear what seasonal variations are meant here. Seasonal variations of environmental conditions that control DOC exports or controls of the seasonal variations of the DOC export itself?

**REPLY:** We meant the seasonal variations of environmental conditions regulating DOC export. The sentence will be modify accordingly:

"To better constraint the seasonal variations of the environmental conditions regulating dissolved organic matter (DOM) export in lowland catchments, we […]"

L8: Why "nitrates" not "nitrate"?

**REPLY:** "Nitrate" will be used in the revised manuscript instead of "nitrates".

L13: "visit" is maybe not the best choice here. I hope you also sampled them.

**REPLY:** Indeed, we "collected" samples.

L14ff: Increase of DOC concentrations and release into the soil water seems to be the same thing. Our do you mean release into surface water?

**REPLY:** We meant an increase in soil waters:

"We observed a large increase in DOC concentrations in soil waters of the riparian areas […]"

L15: I have mixed feelings about "notably due to". Is that your interpretation of the data or a proof? Maybe another choice can make that level of certainty in the underlying processes more clear.

**REPLY:** It is based on the data that show a strong link between iron biodissolution and DOC release in soils. We changed "notably due to" by "linked to".

*Introduction*

L36-39: Consider to state the hydroclimatic conditions under that these statements were made. Also consider the work of Winterdahl in this context (10.1002/2013gb004770).

**REPLY:** Thank you for the additional reference. We suggest modifying the text as follow:

"Numerous research carried out in temperate and boreal catchments have shown that headwater catchments are the main entry point of DOM into fluvial networks (Ågren et al., 2007; Creed et al., 2015) and identified riparian areas as the dominant sources of DOM at the catchment scale owing to their location at the terrestrial-aquatic interface (Sanderman et al., 2009; Lambert et al., 2014; Laudon et al., 2012; Winterdahl et al., 2014)."

L41-42: Is this statement underpinned with papers later in the text? Maybe it make sense to state that time-scale question after the next section? Here it seems quite strong without underpinning.

**REPLY:** This statement is supported by several papers, indeed cited in the following paragraph. Please note that the introduction will be modidy following comments from B. Selle:

"Numerous research carried out in temperate and boreal catchments have shown that headwater catchments are the main entry point of DOM into fluvial networks (Ågren et al., 2007; Creed et al., 2015) and identified riparian areas as the dominant sources of DOM at the catchment scale owing to their location at the terrestrial-aquatic interface (Sanderman et al., 2009; Lambert et al., 2014; Laudon et al., 2012; Winterdahl et al., 2014). The flushing of shallow organic-rich soil layers during storm events typically represents the majority of annual DOC loads (Inamdar et al., 2006), and the DOC versus discharge relationships during storm events show that DOC export is transport-limited at the event scale (Buffam et al., 2001; Zarnetske et al., 2018). Although geomorphological and climatic conditions regulate DOC loads in stream waters (Winterdahl et al., 2014; Laudon et al., 2012), DOC export at the annual scale is commonly conceptualized as a two-steps process in which DOM is produced and stored in the catchment during the hot and dry period, and then exported toward surface waters during the wet and cold period (Boyer et al., 1996). This two-steps process model often described in temperate catchments (Deirmendjian et al., 2018; Strohmenger et al., 2020; Wen et al., 2020; Ruckhaus et al., 2023) is also supported by numerous studies carried out in tropical (Bouillon et al., 2014), boreal (Tiwari et al., 2022), Mediterranean (Butturini and Sabater, 2000) or Arctic fluvial networks (Neff et al., 2006). However the processes regulating the size of the pool of riparian DOM remain unclear (Tank et al., 2018 and references below)."

L43-56: Again some mentioning of climatic settings may help. I noted that often there is the assumption that DOC export works basically similar from boreal to Mediterranean/ subtropic conditions. Partly justified but (not exhaustively) stating where some of the findings have been made would be great. See also Laudon (10.1029/2012gl053033).

**REPLY:** Researches on DOC export have shown that stream DOC export is very similar across different geomorphological and climatic settings. Please find below a proposition of modification:

"Although geomorphological and climatic conditions regulate DOC loads in stream waters (Winterdahl et al., 2014; Laudon et al., 2012), DOC export at the annual scale is commonly conceptualized as a two-steps process in which DOM is produced and stored in the catchment during the hot and dry period, and then exported toward surface waters during the wet and cold period (Boyer et al., 1996). This two-steps process model often described in temperate catchments (Deirmendjian et al., 2018; Strohmenger et al., 2020; Wen et al., 2020; Ruckhaus et al., 2023) is however also supported by numerous studies carried out in tropical (Bouillon et al., 2014), boreal (Tiwari et al., 2022), Mediterranean (Butturini and Sabater, 2000) or Arctic fluvial networks (Neff et al., 2006)."

L67-72: I have problems following the line of argumentation in this sentence. Why is the impact of Fe reduction (on DOC export? Not stated here) limited but then rather favoring conditions are mentioned.

**REPLY :** We reformulated this sentence for clarity:

"However, the triggering of Fe reduction (and the consequences in terms of DOC release in soils and export toward stream waters) could be limited in agricultural catchments owing to inputs of nitrate (an oxidizing specie) from upslope and/or groundwaters that may prevent the reductive biodissolution of iron (Mcmahon and Chapelle, 2008; Christensen et al., 2000)."

L75-77: This is true but it would be fair to cite some studies that make this attempt of bringing together soil and stream water (Knorr 2013, Dupas et al. 2015 – though P-centered, or even Seibert et al. 2009 and Ledesma et al. 2015).

**REPLY:** We will add several papers in addition to those suggested by the reviewer: Knorr, 2013; Dupas et al., 2015; Ledesma et al., 2015; Seibert et al., 2009; Sanderman et al., 2009; Lambert et al., 2013.

L82-91: Maybe mention if and how stream water quality was monitored as well.

**REPLY:** Indeed. The text will be modify as follow:

"To this end, zero-tension lysimeters were installed in the riparian area of a well-instrumented agricultural catchment whose stream waters are continuously monitored for water quality, including DOC at high frequency (Fovet et al., 2018).".

*Material and methods*

L102: Some more details on the riparian soils would be helpful to later on make more clear, why a depths of 15 cm was chosen.

**REPLY:** Values of soil organic carbon content have been added in the revised manuscript:

"Soil organic carbon content presents lateral (riparian versus upland soils) and vertical (surface versus deep soils) gradients, with highest values about 5.3 – 5.6 % reached in the uppermost soil horizons (0-20 cm depth) of the riparian area while soil organic content fall under 1% below 20 cm depth (Lambert et al., 2011)."

L107: Figure reference is misleading here since there is no land use in this figure.

**REPLY:** Correct. Figure will be modify.

L122: Some details on the zero tension lysimeters would be helpful.

**REPLY:** A figure showing how lysimeters are build will be added in supplements, based on Dupas et al. (2015):

[Figure]

L128: Not sure if "degradation" is the right word.

**REPLY:** This will be replaced by "damages".

L129: "consecutive dates of data" sounds a bit strange.

**REPLY:** "of data" will be removed.

L132: Figure 1 reference does not fit here.

**REPLY:** Indeed, we meant Figure 2.

L137 and L146: Remove the XXX.

**REPLY:** Done.

L157: On which bases was the decision made to dilute the sample.

**REPLY:** Samples were diluted following the Ohno et al. (2002) recommendation (absorbance at 254 nm < 0.3) to reduce inner filter effects and iron quenching.

L177: How were missing data handled?

**REPLY:** Missing data were very few (less than 10 sampling) and were not included in the PCA.

L181: Does normalization not include an averaging to a mean of zero and a standard deviation of 1? Is the averaging justified? Temporal variability may be higher than spatial variability and maybe each single sample would be better in the PCA? This needs more justification somewhere in the paper.

**REPLY:** Similar comments were raised by C. Williams. Please find below our answer:

The aim of the PCA-clustering approach was to discriminate and group lysimeters based on the occurrence or absence of iron biodissolution in soil waters in order to investigate the temporal pattern of each cluster that would help to identify patterns compared to individual time series. For this reason, data we aggregated for each lysimeters. Otherwise, a given lysimeter would switch clusters and the temporal figure per cluster would make no sense.

Although we agree that the data aggregation per lysimeter erases the temporal dimension, this is necessary for the clustering and the temporal aspect is described in the next step of the analysis. Please note that including all the dates lead to similar result compared to the "temporally-normalized PCA" used in the manuscript, although, obviously, more 'noisy' (Figure 1).

[Figure]

Figure 1 - PCA using all dates

The revised manuscript gives more details to justify the approach (section 2.5):

"A principal component analysis (PCA) coupled to a clustering analysis was used to discriminate and group lysimeters based on the occurrence or absence of iron biodissolution in soil waters in order to investigate the temporal pattern of each cluster that would help to identify patterns compared to individual time series. For this reason, data (DOC, NO3-, SRP and Fe(II) concentrations and the relative contribution of PARAFAC components) were aggregated for each lysimeters and normalized."

*Results*

L189: In a stricter sense this opener connects your catchment to a general behavior in the region that would actually better fit the discussion and, moreover, needs referencing. I suggest to just state three phases. You may mention the general regional behavior in that type of catchment in the introduction or method section also. Moreover, the sharp mid-months boundaries of the phases need a criterion to be mentioned here. Is this visually decided?

**REPLY:** The different phases are based on the water table fluctuation along the hillslope and were defined by Lambert et al., 2013. Following this comment we proposed to modify the text as below:

"The hydrological regime of the study site is characterized by the succession of three distinct periods determined by water table fluctuations along the hillslope, corresponding to different hydrological regimes for the riparian soils (Fig. 2; Lambert et al., 2013): (i) a period of progressive rewetting of riparian soils after the dry season and of low groundwater flow and low stream discharge (01/09/2022 – 18/12/2022, mean and cumulated precipitation = 5.1±5.3 mm d-1 and 338.5 mm, respectively); (ii) a period of prolonged waterlogging of riparian soils induced by the rise of water table in the upland domain, corresponding to high values of hillslope groundwater flow and stream discharge (18/12/2022 – 9/05/2023, mean and cumulated precipitation = 6.6±7.9 mm d-1 and 577 mm, respectively); and (iii) a period of drainage and progressive drying of the riparian soils induced by the drawdown first in the upland domain then in the bottomland domain and corresponding to the decrease of both the hillslope groundwater flow and stream discharge (09/05/2023 – 01/07/2023, mean and cumulated precipitation = 2.1±4.5 mm d-1 and 23 mm, respectively)."

Fig. 2: Consider to indicate the phases in the three time series.

**REPLY:** Indeed, the figures will be modified.

L197: You mention frequent intense rainfall with 2 mm/d but wrote about moderate precip of 5 mm/d above. Are the numbers correct?

**REPLY:** The 2 mm per day referred to the short period inside the second period during which no significant rainfall events occurred. In the revised manuscript we removed this sentence for clarity, only mentioning the precipitation values for each period.

L201: Why not giving precipitation values for this third period?

**REPLY:** we add the values.

L209ff: The chapter on fluorescence properties does not fit here in my opinion. It feels more naturally for me to first state water quality in terms of concentrations and then jump to the DOM properties in detail. But you may have a good reasoning at hand.

**REPLY:** We found more logical to first begin with the PARAFAC model as the temporal and spatial variations in DOM composition are linked to the dynamics of iron. We thus found sequence PARAFAC > Seasonal Variations > PCA & cluster more relevant than Seasonal Variations > PARAFAC > PCA & cluster.

Fig. 3: Be precise in the captions. What temperature is this? Mention that C-F are soil water concentrations? Homogenize captions and axis – e.g. nitrates – N-NO3? The latter also applies to the other figures.

**REPLY:** The 'soil' mention will be added on axes, and we will homogenize captions and axis by keeping, for instance, N-NO3 rather than nitrate.

Fig. 4: Does it make sense to give nitrate in B on a log axis?

**REPLY:** Some values were equal to 0, as measurements were below detection limit. A log axis would therefore not be adequate.

L231: I suggest to also give a Pearson's correlation coefficient to underline the claim of a linear relationship. However, nitrate-iron relationship seems to be far away from linear and rank correlation would be maybe a better fit.

**REPLY:** Correlation coefficients will be added in the figures and/or in the text. Given the relationship, we will use either Pearson or Spearman correlation coefficients.

L234: "Some" lysimeter not following that cannot be seen in the data. You probably did the correlation across all data. Maybe state lysimeter-individual correlation coefficient (mean, range) to point out that some do not follow the general behavior?

**REPLY:** Details will be added.

L243: This title should make clear that this is about soil water quality.

**REPLY:** This will be modify.

L244: Would be good to write clearly what variance is explained.

**REPLY:** We modified the text as below:

"Overall, the two first components of the PCA explained 69.4 % of the total variance of the dataset"

"discriminating lysimeters depending on the degree of Fe(II) biodissolution" does not sound well written and seems to be more a discussion than a result. I would leave that statement more to the lines 251-253 at the end of the section.

**REPLY:** The sentence was modified as below for clarity, but we still mentioned that lysimeters were discriminated based on occurrence or absence of iron biodissolution. Indeed, we think that this is a result from the PCA.

"[…] discriminated lysimeters depending on the occurrence or absence of Fe(II) biodissolution in soil waters of the riparian area"

L254: In this chapter it may be elaborated on if averaging the lysimeters in time was justified? Does one lysimeter seem to switch from high-nitrate:low-Fe to low-nitrate:high-Fe over time? This is hard to be seen in Fig. 6 as here all lysimeters of one sampling data and one cluster are averaged.

**REPLY:** Please see our previous answer on the PCA approach we used.

Fig. 6: Same as fig. 2 – indicating the different wetness phases would be very helpful here.

**REPLY:** This will be done.

L258 and 270: I don`t understand the idea of a flushing dynamic. Does that mean the solutes are flushed out and replaced by a different water with a different quality? Moreover, I find this to be more an interpretation of that data and thus more a discussion part.

**REPLY:** We agree that this paragraph is unclear and that we used terms more relevant for the discussion. Following this comment, we modified the text as below:

"In the cluster 1, DOC, N-NO3 and SRP decreased from 39.8±13.3 to 23.4±8.4 mg L-1, from 2.6±3.6 to 1.2±1.8 mg L-1, and from 0.18±0.18 to 0.08±0.15 mg L-1, respectively, during the rewetting phase of the catchment while Fe(II) was no measured at significant levels. During the high flow period, however, Fe(II) increased gradually from 3.7±3.2 to 26.5±7.8 mg L-1, and both DOC and SRP followed a similar trend with concentrations raising from 27.3±9.5 to 54.9±25.0 mg L-1 and from 0.07±0.13 to 0.18±0.11 mg L-1, respectively. During this period and until the end of the hydrological cycle, N-NO3 were very low, falling from 0.54±0.66 mg L-1 at the beginning of the high flow period to values below 0.15 mg L-1 the rest of the survey. The start of the third hydrological period corresponding to the drawdown of the water table and the consecutive aeration of riparian soils was marked by the rapid drop of Fe(II) at 8.1±7.4 mg L-1, DOC at 17.5±10.9 mg L-1, and SRP at 0.02±0.02 mg L-1."

Please note that we applied the same kind of modification for the following paragraph where the term flushing was also used:

"Similarly to cluster 1, soil solutions from the cluster 2 exhibited a decline in DOC and SRP concentrations during the rewetting phase of the catchment but these trends continued during the high flow period, with minimal values reached in the middle of February. Thus, DOC dropped from 34.5±7.1 to 9.4±3.1 mg L-1 and SRP from 0.19±0.08 to 0.02±0.01 mg L-1 during this period, before showing an increasing trend to reach concentrations about 21.0±6.1 mg L-1 for DOC and 0.16±0.13 mg L-1 for SRP at the end of the high flow period. DOC remained elevated (24.1±3.1 mg L-1) at the start of the dry period, but SRP dropped close to depletion. To the inverse, N-NO3 first increased from 0.57±0.81 in November to

maximum values of 6.5±5.9 in the middle of March, and then exhibited decreasing concentrations until a complete depletion at the beginning of the third hydrological period. Contrary to cluster 1, Fe(II) was not measured at significant concentrations in cluster 2 (i.e. below 0.5 mg L-1) except in March, during which Fe(II) increased from 1.2±1.9 to 4.1±0.2 mg L-1."

*Discussion*

L290: appearance, abundance maybe better than apparition.

**REPLY:** We replaced by the term 'release' that fit more the purpose.

L303: What about a temperature effect here? Or can rainwater just not infiltrate deep enough?

**REPLY:** Although it is true that temperature may affect oxygen solubility and exchanges with atmosphere, we did not found any relationship between Fe(II) and soil temperatures. The gradual increase in Fe(II) also suggests that rainfall events did not impacted iron biodissolution.

L306f: I don't understand the role of the hydraulic gradient here. Translating to soil water/ shallow groundwater travel time? What I read is more about depth to groundwater than hydraulic gradient.

**REPLY:** The hydraulic gradient determines the flow of groundwater coming from upland domain to the riparian soils, impacting therefore the hydrological functioning of valley bottoms (Lambert et al., 2013). Although we are not certain about the exact mechanism involved, we attributed the slight decrease in Fe(II) in February/March to the slight decrease in water table in the upland domain. The paragraph was modified to take into account this comment and the previous one:

"A fundamental condition for the establishment of reductive conditions is the prolonged waterlogging of riparian soils. As shown earlier for this and other lowland catchments on impervious bedrock, the increase of the hydraulic gradient induced by the rise of groundwater in the upland domain during the high flow period maintains a strong hydrologic connection between upland and riparian domains (Pacific et al., 2010; Molenat et al., 2008). Under these conditions, riparian soils remain waterlogged owing to a high and continuous hillslope groundwater flow, leading to the gradual establishment of reductive conditions and the subsequent triggering of Fe-biodissolution as long as inputs of oxidizing species remained limited and/or counterbalanced by higher rate of consumption through microbial activity (Lotfi-Kalahroodi et al., 2021; Lambert et al., 2013). This pattern was well illustrated by records from lysimeters grouped in the first cluster (Fig. 6). After a quick depletion of an initial stock of nitrate accumulated during the previous summer, reductive conditions were rapidly established at the beginning of the high flow period and increasing Fe(II) concentrations in soil solutions evidenced the triggering of the reductive Fe biodissolution in riparian soils. The gradual increase in Fe(II) during all the high flow period despite variations in temperature or rainfall patterns (with some intense precipitation events > 20 mm d-1) pointed to limited impact, except during a period of low precipitation during which both Fe(II) and DOC exhibited a slight decrease in February/March. Although the exact reason remains to be determined, we attributed this small drop to the drawdown of the water table in upland groundwater flow owing to a prolonged absence of precipitations (see PK3 fluctuations, Fig. 2) that may have favoured the penetration of oxygen within soil waters (as no changes in N-NO3 occurred)."

L307: If DOC and SRP are similarly affected by oxygen and iron presence, why are they weighted differently in the two clusters?

**REPLY:** Likely because SRP concentrations were lower compared to DOC.

L316: Double word biodissolution here.

**REPLY:** Indeed.

L318f: I am not sure if I understand what is meant by "net depletion pattern".

**REPLY:** Both DOC and SRP concentrations decreased in soil waters, that we interpreted as the progressive flushing of a finite DOC/SRP pool.

L327 (and 313): Spatial patterns are not shown but may be helpful and interesting? So maybe map the clusters back to the catchment figure? Could be in the SI or an additional panel in Fig. 5.

**REPLY:** Lysimeters were aligned along three lines parallel to the stream channel. These lines, about 10-20 m from each other, were located at different distance from the stream with the aim to capture the heterogeneity of water flow paths and nitrates concentrations coming from the upland domain. Despite our sampling design, the distance between each lysimeter is not a variable that we could integrate in our analysis. We would need to set distance to an independent point (the nearest field? the river?) but we don't think this is would lead to an interesting pattern as no spatial pattern was visible: two neighbouring lysimeters could be more different than lysimeters on the opposite side of the

Following this comment, more details were added in the Material and methods (section 2.2):

"Lysimeters were aligned along three lines parallel to the stream channel. These lines, about 10-20 m from each other, were located at different distance from the stream with the aim to capture the heterogeneity of water flow paths and nitrates concentrations coming from the upland domain. Lysimeters were all located in the hydromorphic soils unit (Figure 1)."

And in Results (section 3.3):

"Despite the fact that lysimeters were installed along three lines that were more or less closed to the stream channel, no spatial pattern was identified. Thus, two neighbouring lysimeters could be more different than lysimeters on the opposite side of the transect."

L337: Water circulation is maybe not the most precise word here. Does soil water really circulate?

**REPLY:** We modified the sentence:

"A first explanation can be related to the heterogeneity in water flowpaths in soils."

L353-357: This is an interesting part but would, at this process-scale, better fit the end of chapter 4.1? But I am not fully sure either. I noted that this is an initial statement on something explained in more detail below (L373ff). Maybe make more clear that details are given in the following text?

**REPLY:** We think it is a good opening for the 4.2 section. Indeed, the idea behind these sentences (that winter should be considered as an active phase of DOC export) are developed in the next paragraphs where we try to link soil and stream dynamics.

L366-367: I know what you mean here but generally a "supply-limited pool" is associated with a dilution behavior not a flushing behavior. So, check your choice of words here and maybe better describe what concentration dynamics you see at this point in time in the stream.

**REPLY:** We replace by "DOM pool limited in size" which is more relevant.

*Conclusion*

L401: The word "but" indicates some contradiction which I do not clearly see here. Fe-reduction can only establish when nitrates are not present, right?

**REPLY:** Indeed, we should reconsider this sentence. We proposed the following changes:

"In agreement with previous studies (e.g. Selle et al., 2019; Knorr, 2013), the establishment of Fe-reducing conditions in riparian areas was identified as a major mechanism for the release of large amount of DOM in soil waters. In agricultural catchments, however, we found that this process can be buffered by nitrate, leading to a strong heterogeneity on the degree of iron biodissolution and its consequences on soil DOC dynamics. Our study also evidenced that another production mechanisms unrelated to Fe dynamics contributed to release DOM in riparian soils, pointing the need to investigate deeper stream DOC export at the soil/stream interface."

L409: Any reference for the "wetter winter" statement?

**REPLY:** Yes: Strohmenger et al. (2020).

L404ff: This last section in the conclusions seem to make a new story on the relation of the small-scale redox soil processes to the long-term trends. While I appreciate this part I think this may be already part of the introduction and motivation.

**REPLY:** In fact this was part of the motivation, but indeed this was very briefly mentioned in the manuscript lines 82-85. The last paragraph of the introduction will be modify to better appreciate the goal of our study:

"To this end, zero-tension lysimeters were installed in the riparian area of a well-instrumented agricultural catchment, the so-called Kervidy-Naizin catchment, whose stream waters are continuously monitored for water quality, including DOC at high frequency (Fovet et al., 2018). This catchment is located in Brittany (France), a region where stream DOC concentrations exhibited contrasting trends (increasing, decreasing or no trend) over the 2007-2020 period despite similar geomorphological and climatic conditions (Supplementary Fig. S1). The Kervidy-Naizin site for instance shows a weak but significant increase in stream DOC concentrations over the last decades (Strohmenger et al., 2020). In this context, another goal of this study was to explore the hyoothesis that long-term decline in nitrate inputs from agricultural practices (Abbott et al., 2018) may have impacted long-term trends in DOC through a potential impact on iron dynamics in riparian soils."

**References**

Dupas, R., Gruau, G., Gu, S., Humbert, G., Jaffrézic, A., and Gascuel-Odoux, C.: Groundwater control of biogeochemical processes causing phosphorus release from riparian wetlands, Water Research, 84, 307-314, https://doi.org/10.1016/j.watres.2015.07.048, 2015.

Lambert, T., Pierson-Wickmann, A.-C., Gruau, G., Jaffrezic, A., Petitjean, P., Thibault, J.-N., and Jeanneau, L.: Hydrologically driven seasonal changes in the sources and production

mechanisms of dissolved organic carbon in a small lowland catchment, Water Resources Research, 49, 5792-5803, https://doi.org/10.1002/wrcr.20466, 2013.

Strohmenger, L., Fovet, O., Akkal-Corfini, N., Dupas, R., Durand, P., Faucheux, M., Gruau, G., Hamon, Y., Jaffrezic, A., Minaudo, C., Petitjean, P., and Gascuel-Odoux, C.: Multitemporal Relationships Between the Hydroclimate and Exports of Carbon, Nitrogen, and Phosphorus in a Small Agricultural Watershed, Water Resources Research, 56, e2019WR026323, https://doi.org/10.1029/2019WR026323, 2020.

---

## Author Response (AR1)

**Response to Reviewer 1 Clayton Williams**

**General Comments**

Lambert et al.'s "Role of nitrogen and iron biogeochemical cycles on the production and export of dissolved organic matter in agricultural headwater catchments" provides an interesting contribution around possible mechanisms influencing dissolved organic carbon release into streams from agricultural soils. The manuscript provides a useful framework to explain why passive (water flow / discharge) transport of DOC from soils to streams does not work fully as a way to predict DOC concentrations in steam. The study proposes and investigates the role of Fe and NO3 in soils as active regulators of DOC release throughout the hydrological cycle. The study provides a useful contribution to our understanding of environmental regulators of stream DOC concentrations. I also think the results of the study highlight very clearly the heterogeneity of the system, which raises important and novel questions about the cumulative effects of soil-water interactions, material transport, and stream conditions.

I enjoyed reading the manuscript and after reflection have a few clarify questions and comments that I hope if addressed would strengthen the manuscript.

**REPLY:** We thank C. Williams for its positive evaluation of our work.

1. With respect to the results and statistical analyses, I did not understand why data were averaged and then used in the PCA at level of each lysimeter. I think the cluster and PCA approach works well here to reduced noise and find patterns, but based on the temporal patterns and dynamic connections between variables, it would appear to me that using each event by lysimeter would better match the papers intent

**REPLY:** The aim of the PCA-clustering approach was to discriminate and group lysimeters based on the occurrence or absence of iron biodissolution in soil waters in order to investigate the temporal pattern of each cluster that would help to identify patterns compared to individual time series. For this reason, data we aggregated for each lysimeters. Otherwise, a given lysimeter would switch clusters and the temporal figure per cluster would make no sense. Although we agree that the data aggregation per lysimeter erases the temporal dimension, this is necessary for the clustering and the temporal aspect is described in the next step of the analysis. Please note that including all the dates lead to similar result compared to the "temporally-normalized PCA" used in the manuscript, although, obviously, more 'noisy' (Figure 1).

[Figure]

Figure 1 - PCA using all dates

We revised the manuscript to explain the purpose of our approach (lines 208-213):

"A principal component analysis (PCA) coupled to a clustering analysis was used to discriminate and group lysimeters based on the presence or absence of iron biodissolution in soil waters. The aim was to help visualize temporal pattern for each of the two clusters rather than 17 time series if data were plotted for each lysimeter. For this reason, data (DOC, NO3-, SRP and Fe(II) concentrations and the relative contribution of PARAFAC components) were averaged for each lysimeters then normalized."

1. Should flow path and spatial network be accounted for? Perhaps proximity of lysimeters to each other is important to explaining pattern?

**REPLY:** Lysimeters were aligned along three lines parallel to the stream channel. These lines, about 10-20 m from each other, were located at different distance from the stream with the aim to capture the heterogeneity of water flow paths and nitrates concentrations coming from the upland domain. Despite our sampling design, the distance between each lysimeter is not a variable that we could integrate in the PCA. We would need to set distance to an independent point (the nearest field? the river?) but we don't think this is would lead to an interesting pattern as no spatial pattern was visible: two neighbouring lysimeters could be more different than lysimeters on the opposite side of the transect.

Following this comment, more details were added in the Material and methods (lines 139-143):

"We placed the lysimeters along three lines parallel to the stream channel, about 10-20 m apart from each other and from the stream, with the aim to capture the heterogeneity of water flow paths and nitrate concentration coming from the upslope cultivated fields. Lysimeters were all located in the hydromorphic soils unit according to the soil map (Figure 1)."

And in Results (lines 260-261):

"Despite the fact that lysimeters were installed along three lines ranging 10-30 m from the stream, no spatial pattern was identified."

2. I agree with the use of PARAFAC components as the sole DOM identifier and not using indices or peaks. The peak shape for each component looks normal, but I was surprised that the PARAFAC model did not include a protein-like peak. This seems odd and I can't think of a study using PARAFAC that lacks some version of a protein-like peak even if contained within a multi-peak component. I am not certain what to make of this. Usually humic-like peaks dominant soils but protein-like peaks are still present and on an absolute basis can contain more protein-like materials than what would be found in a stream. Perhaps, the original uncorrected EEMs, blanks, and corrections could be revisited and verify that the model correctly represents their features. Perhaps including a few corrected-observed, modeled, and residual eem plots would be useful to highlight that protein-like peaks were not present. This way the reader can be reassured the model fit well the data. Assuming the absence of a protein-like component, I think this result needs to be discussed and clarified. Protein-like and less complex DOM forms are expected to be present in agricultural catchments and their absence would be interesting to explore more deeply in the discussion.

**REPLY:** We agree with C. Williams that protein-like components are commonly reported in PARAFAC models built in surface and soil waters. However, we can exclude any bias from measurement and/or from our modelling approach. First, a blank was systematically measured before samples to verify instrument calibration (e.g. Raman peak position) and noise. Second, several steps were rigorously controlled before attempting the final validation of the model including 1) the comparison of measured *versus* corrected EEM (corrections including blank subtraction, inner filter effect…as described in the Material and Methods section), 2) the randomness of residual EEMs, and 3) the correct aspect of PARAFAC components. No peaks could be visualized nor identified at low Ex/Em wavelengths all along the modelling process, and no solution proposed by PARAFAC included any protein-like component (tests were realized with a number of components ranging from 4 to 7).

That being said, we know from previous studies that soil DOM can include protein-like components.

We provided explanations in a new paragraph (lines 459-479):

"The PARAFAC results suggest that DOM mobilized from soil to streams is only composed by aromatic molecules of high molecular weight. Although complex organic molecules indeed dominate stream DOM export (Fellman et al., 2009), it should be noted however that protein-like components are commonly found in stream waters (Inamdar et al., 2012), including in

our study site (Humbert et al., 2020). The lack of such components in our model results from our sampling approach and not from their absence in catchment soils. Indeed, the production of protein-like components in catchment soils is restricted to the summer hot and dry period during which a pool made of low-aromatic and microbially-derived compounds built up in riparian soils (Lambert et al., 2013). However, this DOM pool is quickly flushed and exhausted during the rewetting phase in October-November, and soil DOM during the winter period is mainly composed by highly-aromatic molecules originating from soil organic material (Lambert et al., 2014). Agricultural practices such as fertilizer applications can represent another source of protein-like DOM in the catchment (Humbert et al., 2020), but these inputs remain episodic with a low impact on DOM at the catchment scale (Humbert et al., 2015; Lambert et al., 2014). For instance, a recent one-year of monitoring of soil waters at different locations in the catchment has shown that protein-like components represent only 3.44 ± 2.8% of the total fluorescence signal in catchment soils, this contribution being particularly low in riparian areas (Humbert et al., 2020). Therefore, the absence of protein-like components in our PARAFAC model is the consequence of our sampling design that focused on DOM production mechanisms in riparian soils (distant from agricultural inputs) during the winter period (period of production of highly aromatic compounds in soils)."

3. I think the connection between DOM quality and Fe-NO3 interactions could be explored more fully. The PCA results are mostly explored from a Fe, DOC, and NO3 point of view, but DOM seems to also be divided along the two clusters. I think more could be done to amplify the DOM spilt along PC1 within the two clusters. The discussion pulls in DOM quality as a possible mechanism and I think some of these links could be brought out more in the results. Perhaps variable influence scores for the PCA showed Fe, DOC, and NO3 were most important and this is why DOM was only partly included as a clustering agent. If so, I think it would be useful to the reader to acknowledge these influences.

**REPLY:** We agree that the discussion in its current form is relatively poor regarding DOM dynamics. Following this comment, we investigated a little bit more our results to explore deeper the interactions between DOM quality and Fe-NO3 interactions with a focus on PARAFAC component C4 that presented the most interesting and interpretable pattern. This lead to the following changes.

Results (lines 283-291): "PARAFAC components had similar or even higher scores than DOC, Fe(II), and NO3 concentrations on the two first dimensions of the PCA (Supplementary Fig. S2), illustrating the importance of DOM composition as an important factor contributing to explain the spatial variability across lysimeters. The distribution of PARAFAC components along the first dimension reflects the relationships between their relative contribution and Fe(II), concentrations (not shown). More specifically, %C4 was strongly and positively correlated with Fe(II) ($R^2$ = 0.38, Pearson r = 0.62) compared to other components that exhibited weakest and negative relationships with Fe(II) ($R^2$ from 0.09 to 0.19, Pearson r from -0.30 to -0.43)."

Discussion (lines 391-404): "The PARAFAC components identified in the model suggest a dominance of highly aromatic and conjugated molecules across all lysimeters and dates, which is typical of DOM derived from soil organic matter and found in poorly drained soils in riparian or wetland areas (Sanderman et al., 2009; Lambert et al., 2013; Yamashita et al., 2010). The larger proportion of C4 in the first cluster however indicates that the Fe oxyhydroxides reduction leads to greater proportion of microbially-derived compounds within the DOM pool. In agreement with previous studies showing that the Fe(III) reduction could enhance the decomposition of organic matter in soils (Chen et al., 2020; Kappler et al., 2021), the close link between Fe(II) and C4 likely reflects an indirect effect of Fe biodissolution promoting the degradation of soil OM and the subsequent incorporation of microbially-derived compounds into the DOM pool (Dong et al., 2023). This hypothesis is well consistent with previous experimental studies performed with soils from the Kervidy-Naizin riparian area, which showed that bacterial reduction of Fe(III)-oxides to Fe(II) was concomitant with the release of large biological organic by-products upon the growth of bacterial communities (Lotfi-Kalahroodi et al., 2021)."

[Figure]

Supplementary Figure S2 – Matrix of coefficient correlation between variables and the dimensions of the PCA.

4. The discussion did an excellent job connecting patterns and telling a story around the variables important to the release of DOC from streams into soils. I struggled a little seeing some of these patterns in the figures and I wasn't always certain how to interpret the pattern within PCA clusters. I think it would be helpful to include a little more detail in the results that explains specific patterns visualized in the figures. This then could be revisited in the more detail already present in the discussion. In addition, perhaps there might be an X vs Y type approach that could be used to amplify the patterns expressed in the timeseries plots.

**REPLY:** After considering this comment we recognize that our data do not fully support the statements we made in the discussion trying to link the stream DOC dynamics to what we observed in riparian soils. Although we know from previous studies that riparian soils where we installed our lysimeters are the dominant source of DOM at the catchment scale, we were limited in our ability to link stream DOC dynamics to the patterns identified in the clusters. Linking soil and stream DOC would have required a common and robust tracer, as we have done previously in this catchment using stable carbon isotopes (Lambert et al., 2013; Lambert et al., 2014).

The manuscript was modify accordingly:

Lines 424-438: "However, the high-frequency measurements of DOC in the stream do not fully support this statement. The establishment of a hydrological connection between riparian soils and the stream during the winter period showed the stream DOC to gradually decrease both at peak discharge during successive storm events and at base flow during inter-storm periods (Figure 5). This pattern, which repeats every year in this catchment (Strohmenger et al., 2020), is well consistent with the hypothesis of the mobilisation and exhaustion of a DOM pool limited in size built during the summer period (Humbert et al., 2015). However, stream DOC were found to increase slightly in March/April after the low-flow period that showed the hydrological connection between soils and the stream to decrease. It is unlikely that the mobilisation of an additional pool of DOM from upland soils may explain this small raises in

stream DOC because this pool is 1) relatively small in terms of size, and 2) quickly exhausted at the beginning of the winter period (Lambert et al., 2014). Therefore, the seasonal pattern of stream DOC likely reflects the regeneration of the riparian DOM pool during the winter period as shown by our data collected in soil waters of riparian wetlands."

Lines 439-458: "Stable carbon isotopes have indeed demonstrated that riparian soils of the Kervidy-Naizin catchment – and more particularly the DOM-rich uppermost soil horizons – are the dominant source of stream DOC at the catchment scale (Lambert et al., 2014), a feature commonly shared by headwater catchments (e.g. Sanderman et al., 2009). Thus, the decline in DOC and SRP observed in soil waters, particularly in the second cluster whereby these elements became almost depleted (Fig. 7), was consistent with the general flushing behaviour of the catchment shown by stream DOC from November to February. Similarly, the large two to three fold increase in DOC concentrations in riparian soils (in cluster 1 and 2, respectively) denotes a large mobilisation of DOM between March and May despite wet and low temperature conditions, that could explain in turn the pattern observed in stream DOC at the same time. While part of this regeneration can be attributed to iron biodissolution, the release of large amount of DOC the cluster 2 where the reductive biodissolution of Fe(III) was limited implies that another production mechanisms contributed to release DOM in riparian soils. It is unlikely that agricultural inputs (crop residues, manure application, etc) main may explain the increases in the riparian area, as these sources are episodic and/or size-limited (Lambert et al., 2014; Humbert et al., 2015; Pacific et al., 2010). This observation echoes previous works on the Kervidy-Naizin catchment showing effective interannual regeneration mechanisms of the pool of soluble phosphorus in soils unrelated to iron dynamics (Gu et al., 2017), a statement supported here by the fact that SRP concentrations followed a similar pattern as DOC in soils grouped in the second cluster (Fig. 7)."

Lines 480-492: "Taking together, our results have two important implications regarding our conceptualisation of DOM export in headwater catchments. First, it challenges the idea that the wet period acts solely as a passive export period for DOC, with no or little DOC production (Strohmenger et al., 2020; Ruckhaus et al., 2023; Wen et al., 2020). Second, it emphasis that stream DOC dynamics at the outlet is an integrative signal, potentially masking the high spatial heterogeneity of the system owing to complex interactions between biogeochemical cycles in soils, nutrient transfer at the soil/stream interface and hydrological functioning of catchments. While the patterns of stream DOC were consistent with that observed in soils, our study remains however limited in its capacity to quantify the relative contribution of the cluster identified to stream DOC export. Additionally, we do not have the necessary data such as isotopes or molecular markers to elucidate the precise origin and DOM (and SRP) release in soils unrelated to iron biodissolution, and this should be the focus of future work combining experimental and field studies."

The figure 7 (number 5 in the revised manuscript) has been modify and presents now only DOC at the outlet.

[Figure]

Figure 5 – Variation of DOC concentrations (black circles) in the stream based on high-frequency measurements.

**Specific Comments**

**Introduction**

The introduction framed the study and the key background ideas needed to understand the results really well.

**REPLY:** Thank you.

**Methods**

Figure 1: It seems like it would be useful to have lysimeter locations also included on the map. One of my questions was if there are spatial-correlations hidden within the clusters or across clusters that might be able to be explained by looking at location as a variable. Adding the lysimeter locations to the map, would at least help the reader see if spatial patterns seem relevant or not. The paper said some lysimeters of opposite clusters were within 1 m of each other. So their might not be a spatial pattern, but it might be useful to note this or acknowledge more fully the location of each lysimeter.

**REPLY:** As we mentioned in the first main comment, there was no spatial pattern. Please see our answer for more details.

PARAFAC modeling data: It is true that PARAFAC is not influenced by Fe but the optical conditions of EEMs are impacted by iron levels. It might be useful to provide the range of estimated carbon and Fe(II) levels in the diluted sample. This way they could be used to assure the reader that Fe levels are below interference levels.

**REPLY:** The degree of quenching due to the presence of iron varies greatly between samples depending on the iron:DOC ratio but also on DOM composition (Jia et al., 2021; Poulin et al., 2014). In our study, the Fe(II):DOC ratio was 0.30±0.24, which was in the upper range of the study by Poulin et al. (2014) but very low compared to the study by Jia et al. (2021) that investigated the effect of Fe(II) on fluorescence properties of DOM from an anaerobic aquifer containing up to 300 mg L$^{-1}$ Fe(II), with Fe(II):DOC ratio up to 7. It should be noted however that Fe(III) also impacts DOM fluorescence (Ohno et al., 2008), limiting our ability to quantify the effect of iron on quenching.

That being said, quenching was clearly apparent in some samples (n < 10) that showed the fluorescence intensity to increase with dilution factor, reflecting the influence of high level of Fe that reduces DOM fluorescence (Poulin et al., 2014). The quenching was mainly affecting EEMs at low (< 270) and moderate to high (420 - 490) excitation and emission wavelengths, respectively, which is consistent with previous studies concluding that Fe mainly impacts fluorescence intensity in EEM locations associated with humic-like fluorophores, namely A and C peaks (Jia et al., 2021; Poulin et al., 2014). Thus, although we cannot rule out an effect of iron on EEMs, this would have impacted humic-like fluorophores associated with C1 and C2 components of our model. Considering that these components behaved similarly between clusters and across the season (Figure 4), we argue that Fe would have a limited impact on the conclusion of our study regarding DOM dynamics.

We added a paragraph in the Material and Methods section (lines 178-192):

"In our study, the Fe(II):DOC ratio was 0.30±0.24, implying that significant interferences on DOM fluorescence from iron can be expected (Poulin et al., 2014). The degree of iron quenching, however, varies greatly between samples depending on the iron:DOC ratio (Pullin et al., 2007) but also on DOM composition (Jia et al., 2021; Poulin et al., 2014) and Fe(III) concentrations (Ohno et al., 2008), making difficult to predict the influence of Fe on EEMs. That being said, quenching was clearly apparent in some samples (n < 10) that showed the fluorescence intensity to increase with dilution factor, reflecting the influence of high level of Fe that reduces DOM fluorescence (Pullin et al., 2007). The quenching impacted EEMs at low (< 270 nm) and moderate to high (420 – 490 nm) excitation and emission wavelengths, respectively, which is consistent with previous studies concluding that Fe mainly impacts fluorescence intensity in EEM locations associated with humic-like

fluorophores, namely A and C peaks (Jia et al., 2021; Poulin et al., 2014). Thus, although we cannot rule out an effect of iron on EEMs, this would have impacted the relative contribution of humic-like fluorophores associated with C1 and C2 components of our model (see below) who behaved similarly between clusters and across seasons."

With respect to absorbance scans, would it be possible to clarify if scans were made on filtered or filtered and diluted soil-water samples? High levels of Fe and NO3 have been suggested to impact absorbance scans towards the UV end of the spectrum.

**REPLY:** Absorbance scans were made on filtered and diluted samples. The only purpose of these measurements were to correct EEMs for inner filter effect, we did not use them for the calculation of indices for DOM composition (e.g. SUVA, slope ratio,…) due to the effect of iron and nitrates as mentioned by C. Williams. Note that the inner filter effect corrections were small, as absorbance spectra were furthermore divided by the dilution factor applied for FDOM measurements.

Lines 170-172: "Samples were diluted in most case due the DOM-rich nature of soil waters. The only purpose of CDOM spectra was to correct excitation-emission matrices (EEMs) for inner filter effects (Ohno, 2002)."

Line180-181: Data average then normalized by lysimeter. Does this mean temporal resolution was lost? Please explain the reasoning for this. This collapses the data to n = 17 with 9 variables in multivariate analysis, which could impact how clusters and relationships form. Why not run a PCA on all data and then determine the centroid and error for each lysimeter and build clusters from the full dataset? By averaging, the link between variables within the analysis seems to be broken on a case-by-case basis. It seems in order to say at each event that Fe, DOC, and NO3 changed together, then the PCA should be able to form components with these connections available on a per event basis rather than across sampling location average for the year.

**REPLY:** Please see our answer to the first comment.

Figure 4 suggests correlations used but these analyses and purpose are not reported in methods. Could this approach be added to the methods and correlation coefficients be added to results?

**REPLY:** We agree with these suggestions. Details have been added in the Material and Methods section (lines 217-219):

"Relationships between variables were investigated either through Pearson or Spearman correlations depending of the nature (linear or not) of the correlations."

And presented in the Results (lines 264-267):

"Overall, these elements were strongly linked to each other (Fig. 4). DOC concentrations ranged from 2.3 to 87.4 mg L-1 (mean = 30.2±12.8 mg L-1) over the study period and were linearly and positively (Pearson r = 0.73, p value < 0.0001) associated with Fe(II) that ranged from 0 to 45.8 mg L-1 (mean = 9.8±7.6 mg L-1). Fe(II) was negatively (Spearman r = -0.56, p value < 0.0001) correlated with NO3 (from 0 to 16.4 mg L-1, mean = 0.9±1.1 mg L-1), and SRP (from 0 to 0.5 mg L-1, mean = 0.1±0.1 mg L-1) was also positively (Pearson r = 0.21, p value = 0.0005) related to Fe(II), but not as strongly as for DOC."

**Results**

When exploring the PCA clusters, I think it would be useful to include more emphasis on DOM patterns or provide the variable influence scores that demonstrate that DOM is not a key influence on the clusters

**REPLY:** In fact DOM has a large impact on the PCA. Please see our answer to the first comment.

I am not certain I understand Figure 7 or logic in Figure 6. The discussion suggests that the pattern highlights the role of heterogeneity in soils. Perhaps this idea could be linked here so that it's clear that the patterns don't match. Adding discharge or rainfall data to this figure

might also make it easier for the reader to see the argument in the discussion around passive and active transport.

**REPLY:** This part of the manuscript has been revised. Please see our corresponding answer.

**Discussion:**

L369-372: This is a really interesting idea that very local active processes within a watershed can keep the collective view in a stream relatively constant even though the soil-water patterns are more dynamic. I wonder if there might be a way to tease this idea further out into the results so that it is clear how this idea connects to the broad level patterns in Figure 7 and the very messy lysimeter by lysimeter patterns. I wonder further how much DOC releases from a set of cluster 1 vs cluster 2 like soils is needed to maintain the DOC stream pattern once discharge (passive transport) is accounted for? Would it be possible to roughly estimate load from each cluster as a mixing model with two end members?

**REPLY:** C. Williams raises a very interesting point that is the possibility to quantify the relative contribution of the two clusters to stream DOC export. However, we regret that we cannot address this comment with the current dataset. In previous studies we used stable carbon isotopes (d13C) to quantify the relative contribution of riparian soils to stream DOC export (Lambert et al., 2014) thanks to natural vertical and lateral gradients in d13C values. However, it is likely that DOM from cluster 1 and 2 have similar isotopic signatures, considering that variations in d13C are relatively similar in the upper most soil horizons in the riparian area. Therefore, we would have difficulties in isolating, based on a end-member approach, the two clusters. A modelling approach would be more relevant.

**Technical Corrections**

**REPLY:** We thank C. Williams for noting these technical corrections.

Two periods line 105

(XXX) line 137 after probe name

Line 235: "followed" should be "follow"

Line 244 % with space before number

L316: biodissolution repeated twice

Figure 1 caption, please also explain what PK3 represents.

**References**

Humbert, G., Jaffrezic, A., Fovet, O., Gruau, G., and Durand, P.: Dry-season length and runoff control annual variability in stream DOC dynamics in a small, shallow groundwater-dominated agricultural watershed, Water Resources Research, 51, 7860-7877, https://doi.org/10.1002/2015WR017336, 2015.

Humbert, G., Parr, T. B., Jeanneau, L., Dupas, R., Petitjean, P., Akkal-Corfini, N., Viaud, V., Pierson-Wickmann, A.-C., Denis, M., Inamdar, S., Gruau, G., Durand, P., and Jaffrézic, A.: Agricultural Practices and Hydrologic Conditions Shape the Temporal Pattern of Soil and Stream Water Dissolved Organic Matter, Ecosystems, 23, 1325-1343, 10.1007/s10021-019-00471-w, 2020.

Jia, K., Manning, C. C. M., Jollymore, A., and Beckie, R. D.: Technical note: Effects of iron(II) on fluorescence properties of dissolved organic matter at circumneutral pH, Hydrol. Earth Syst. Sci., 25, 4983-4993, 10.5194/hess-25-4983-2021, 2021.

Lambert, T., Pierson-Wickmann, A.-C., Gruau, G., Jaffrezic, A., Petitjean, P., Thibault, J.-N., and Jeanneau, L.: Hydrologically driven seasonal changes in the sources and production mechanisms of dissolved organic carbon in a small lowland catchment, Water Resources Research, 49, 5792-5803, https://doi.org/10.1002/wrcr.20466, 2013.

Lambert, T., Pierson-Wickmann, A. C., Gruau, G., Jaffrezic, A., Petitjean, P., Thibault, J. N., and Jeanneau, L.: DOC sources and DOC transport pathways in a small headwater catchment as revealed by carbon isotope fluctuation during storm events, Biogeosciences, 11, 3043-3056, 10.5194/bg-11-3043-2014, 2014.

Ohno, T., Amirbahman, A., and Bro, R.: Parallel Factor Analysis of Excitation–Emission Matrix Fluorescence Spectra of Water Soluble Soil Organic Matter as Basis for the Determination of Conditional Metal Binding Parameters, Environmental Science & Technology, 42, 186-192, 10.1021/es071855f, 2008.

Poulin, B. A., Ryan, J. N., and Aiken, G. R.: Effects of Iron on Optical Properties of Dissolved Organic Matter, Environmental Science & Technology, 48, 10098-10106, 10.1021/es502670r, 2014.

**Response to Reviewer 2**

**General comments**

This manuscript presents results from a sampling campaign in the riparian zone shallow groundwater and draws conclusions on the redox conditions influencing DOM exports into the stream of a small agricultural catchment. This work builds on preceding works in the same catchment and extents the previous findings and hypotheses. The manuscript is a good match for HESS and should be of interest for scientists working on catchment water quality.

The manuscript is written in a concise way. Figures are mostly good and references up to date. My specific comments given below add up to a quite long list but are not substantial. My most critical point is the temporal averaging of lysimeter data that nees a better justification. However, from my point of view some work is needed to bring this manuscript into a final acceptable form.

**REPLY:** We thank the reviewer for their positive evaluation of our work.

**Specific comments**

*Abstract*

The abstract uses DOC while title and manuscript text uses DOM. Homogenize that?

**REPLY:** Done.

L7: Check this first sentence. Not clear what seasonal variations are meant here. Seasonal variations of environmental conditions that control DOC exports or controls of the seasonal variations of the DOC export itself?

**REPLY:** We meant the seasonal variations of environmental conditions regulating DOC export. The sentence will be modify accordingly (lines 7-8):

"To better understand the seasonal variations in environmental conditions regulating dissolved organic matter (DOM) export in headwater catchments, we […]"

L8: Why "nitrates" not "nitrate"?

**REPLY:** "Nitrate" is now used in the revised manuscript instead of "nitrates".

L13: "visit" is maybe not the best choice here. I hope you also sampled them.

**REPLY:** Indeed, we "collected" samples.

L14: Increase of DOC concentrations and release into the soil water seems to be the same thing. Our do you mean release into surface water?

**REPLY:** We meant an increase in soil waters (line 14):

"We observed a large increase in DOC concentrations in soil waters […]"

L15: I have mixed feelings about "notably due to". Is that your interpretation of the data or a proof? Maybe another choice can make that level of certainty in the underlying processes more clear.

**REPLY:** It is based on the data that show a strong link between iron biodissolution and DOC release in soils. We changed "notably due to" by "linked to".

*Introduction*

L36-39: Consider to state the hydroclimatic conditions under that these statements were made. Also consider the work of Winterdahl in this context (10.1002/2013gb004770).

**REPLY:** Thank you for the additional reference. We suggest modifying the text as follow (lines 32-36):

"Numerous research carried out in temperate and boreal regions have shown that headwater catchments are the main entry point of DOM into fluvial networks (Ågren et al., 2007; Creed et al., 2015) and identified riparian areas as the dominant sources of DOM at the catchment scale owing to their location at the terrestrial-aquatic interface (Sanderman et al., 2009; Lambert et al., 2014; Laudon et al., 2012; Winterdahl et al., 2014)."

L41-42: Is this statement underpinned with papers later in the text? Maybe it make sense to state that time-scale question after the next section? Here it seems quite strong without underpinning.

**REPLY:** This statement is supported by several papers, indeed cited in the following paragraph. We modified as follow (lines 49-50):

"However the processes regulating the size of the pool of riparian DOM remain unclear (Tank et al., 2018 and references below)."

L43-56: Again some mentioning of climatic settings may help. I noted that often there is the assumption that DOC export works basically similar from boreal to Mediterranean/ subtropic conditions. Partly justified but (not exhaustively) stating where some of the findings have been made would be great. See also Laudon (10.1029/2012gl053033).

**REPLY:** Researches on DOC export have shown that stream DOC export is very similar across different geomorphological and climatic settings. The manuscript was modify lines 40-50:

"Although geomorphological and climatic conditions regulate DOC loads in aquatic ecosystems (Winterdahl et al., 2014; Laudon et al., 2012), DOC export at the annual scale is commonly conceptualized as a two-steps process in which DOM is produced and stored in the catchment during the hot and dry period, and then exported toward surface waters during the wet and cold period (Boyer et al., 1996). This two-steps conceptual model often described in temperate catchments (Deirmendjian et al., 2018; Strohmenger et al., 2020; Wen et al., 2020; Ruckhaus et al., 2023) is also supported by numerous studies carried out in tropical (Bouillon et al., 2014), boreal (Tiwari et al., 2022), Mediterranean (Butturini and Sabater, 2000) or Arctic fluvial networks (Neff et al., 2006). However the processes regulating the size of the riparian DOM pool remain unclear (Tank et al., 2018 and references below)."

L67-72: I have problems following the line of argumentation in this sentence. Why is the impact of Fe reduction (on DOC export? Not stated here) limited but then rather favoring conditions are mentioned.

**REPLY :** We reformulated this sentence for clarity (lines 71-75):

"However, the onset of Fe reducing conditions and the subsequent DOM release could be limited in agricultural catchments owing to large inputs of nitrate (an oxidizing specie) from upslope via groundwater that may prevent Fe reductive biodissolution (Mcmahon and Chapelle, 2008; Christensen et al., 2000)."

L75-77: This is true but it would be fair to cite some studies that make this attempt of bringing together soil and stream water (Knorr 2013, Dupas et al. 2015 – though P-centered, or even Seibert et al. 2009 and Ledesma et al. 2015).

**REPLY:** We will add several papers in addition to those suggested by the reviewer (lines 79-82):

" We still lack studies investigating how processes occurring in soil waters reflect our conceptualization of solutes dynamics based on observations made in surface waters (Knorr, 2013; Dupas et al., 2015; Ledesma et al., 2015; Seibert et al., 2009; Sanderman et al., 2009; Lambert et al., 2013). "

L82-91: Maybe mention if and how stream water quality was monitored as well.

**REPLY:** Indeed (lines 85-87): "To this end, we installed zero-tension lysimeters in the riparian area of the Kervidy-Naizin catchment, whose stream waters are continuously monitored for water quality, including DOC at high frequency (Fovet et al., 2018)"

*Material and methods*

L102: Some more details on the riparian soils would be helpful to later on make more clear, why a depths of 15 cm was chosen.

**REPLY:** Values of soil organic carbon content have been added in the revised manuscript (lines 113-116):

"Soil organic carbon content presents lateral (riparian versus upland soils) and vertical (surface versus deep soils) gradients, with highest values about 5.3 – 5.6 % in the uppermost soil horizons (0-20 cm depth) of the riparian area while soil organic content drop under 1% below 20 cm depth (Lambert et al., 2011)."

L107: Figure reference is misleading here since there is no land use in this figure.

**REPLY:** Correct. Figure has been modified:

[Figure]

L122: Some details on the zero tension lysimeters would be helpful.

**REPLY:** A figure showing how lysimeters are build has been added in supplements, based on Dupas et al. (2015).

Text was modified lines 136-138: "We investigated the seasonal variability in riparian DOM concentration and composition using zero-tension lysimeters designed to collect free soil waters (Supplementary Fig. S2) and […]"

[Figure]

Supplementary Fig. S2

L128: Not sure if "degradation" is the right word.

**REPLY:** This was replaced by "damages".

L129: "consecutive dates of data" sounds a bit strange.

**REPLY:** "of data" has been removed.

L132: Figure 1 reference does not fit here.

**REPLY:** Indeed, we meant Figure 2.

L137 and L146: Remove the XXX.

**REPLY:** Done.

L157: On which bases was the decision made to dilute the sample.

**REPLY:** Samples were diluted following the Ohno et al. (2002) recommendation (absorbance at 254 nm < 0.3) to reduce inner filter effects.

L177: How were missing data handled?

**REPLY:** Missing data were very few (less than 10 sampling) and were not included in the PCA.

L181: Does normalization not include an averaging to a mean of zero and a standard deviation of 1? Is the averaging justified? Temporal variability may be higher than spatial variability and maybe each single sample would be better in the PCA? This needs more justification somewhere in the paper.

**REPLY:** Similar comments were raised by C. Williams. Please find below our answer:

The aim of the PCA-clustering approach was to discriminate and group lysimeters based on the occurrence or absence of iron biodissolution in soil waters in order to investigate the temporal pattern of each cluster that would help to identify patterns compared to individual time series. For this reason, data we aggregated for each lysimeters. Otherwise, a given lysimeter would switch clusters and the temporal figure per cluster would make no sense. Although we agree that the data aggregation per lysimeter erases the temporal dimension, this is necessary for the clustering and the temporal aspect is described in the next step of the analysis. Please note that including all the dates lead to similar result compared to the "temporally-normalized PCA" used in the manuscript, although, obviously, more 'noisy' (Figure 1).

[Figure]

Figure 2 - PCA using all dates

The revised manuscript gives more details to justify the approach (lines 208-213):

"A principal component analysis (PCA) coupled to a clustering analysis was used to discriminate and group lysimeters based on the presence or absence of iron biodissolution in soil waters. The aim was to help visualize temporal pattern for each of the two clusters rather than 17 time series if data were plotted for each lysimeter. For this reason, data (DOC, NO3-, SRP and Fe(II) concentrations and the relative contribution of PARAFAC components) were averaged for each lysimeters then normalized."

*Results*

L189: In a stricter sense this opener connects your catchment to a general behavior in the region that would actually better fit the discussion and, moreover, needs referencing. I suggest to just state three phases. You may mention the general regional behavior in that type of catchment in the introduction or method section also. Moreover, the sharp mid-months boundaries of the phases need a criterion to be mentioned here. Is this visually decided?

**REPLY:** The different phases are based on the water table fluctuation along the hillslope and were defined by Lambert et al., 2013. The manuscript has been modified lines 222-234:

"The hydrological regime of the study site is characterized by a succession of three distinct periods determined by water table fluctuations along the hillslope, corresponding to different hydrological regimes for the riparian soils (Fig. 2; Lambert et al., 2013): (i) a period of progressive rewetting of riparian soils after the dry season and of low groundwater flow and low stream discharge (01/09/2022 – 18/12/2022, mean and cumulated precipitation = 5.1±5.3 mm d-1 and 338.5 mm, respectively); (ii) a period of prolonged waterlogging of riparian soils induced by the rise of the water table in the upland domain, corresponding to high values of hillslope groundwater flow and stream discharge (18/12/2022 – 9/05/2023, mean and cumulated precipitation = 6.8±7.9 mm d-1 and 573 mm, respectively); and (iii) a period of drainage and progressive drying of the riparian soils induced by the drawdown first in the

upland domain then in the bottomland domain and corresponding to the decrease of both the hillslope groundwater flow and stream discharge (09/05/2023 – 01/07/2023, mean and cumulated precipitation = 4.3±4.4 mm d-1 and 42.5 mm, respectively)."

Fig. 2: Consider to indicate the phases in the three time series.

**REPLY:** Indeed, the figures has been modified.

L197: You mention frequent intense rainfall with 2 mm/d but wrote about moderate precip of 5 mm/d above. Are the numbers correct?

**REPLY:** The 2 mm per day referred to the short period inside the second period during which no significant rainfall events occurred. In the revised manuscript we removed this sentence for clarity, only mentioning the precipitation values for each period.

L201: Why not giving precipitation values for this third period?

**REPLY:** We added the values.

L209: The chapter on fluorescence properties does not fit here in my opinion. It feels more naturally for me to first state water quality in terms of concentrations and then jump to the DOM properties in detail. But you may have a good reasoning at hand.

**REPLY:** We found more logical to first begin with the PARAFAC model as the temporal and spatial variations in DOM composition are linked to the dynamics of iron. We thus found sequence PARAFAC > Seasonal Variations > PCA & cluster more relevant than Seasonal Variations > PARAFAC > PCA & cluster.

Fig. 3: Be precise in the captions. What temperature is this? Mention that C-F are soil water concentrations? Homogenize captions and axis – e.g. nitrates – N-NO3? The latter also applies to the other figures.

**REPLY:** Figures have been modified accordingly.

Fig. 4: Does it make sense to give nitrate in B on a log axis?

**REPLY:** Some values were equal to 0, as measurements were below detection limit. A log axis would therefore not be adequate.

L231: I suggest to also give a Pearson's correlation coefficient to underline the claim of a linear relationship. However, nitrate-iron relationship seems to be far away from linear and rank correlation would be maybe a better fit.

**REPLY:** Correlation coefficients has been added in the text (lines 264-268). Given the relationship (linear or not), we used either Pearson or Spearman correlation coefficients (lines 217-219).

L234: "Some" lysimeter not following that cannot be seen in the data. You probably did the correlation across all data. Maybe state lysimeter-individual correlation coefficient (mean, range) to point out that some do not follow the general behavior?

**REPLY:** This sentence was removed in the revised manuscript.

L243: This title should make clear that this is about soil water quality.

**REPLY:** The title is now "3.4 Clustering of soil waters"

L244: Would be good to write clearly what variance is explained.

**REPLY:** We modified the text line 276:

"The first two components of the PCA explained 69.4 % of the total variance of the data […]"

"discriminating lysimeters depending on the degree of Fe(II) biodissolution" does not sound well written and seems to be more a discussion than a result. I would leave that statement more to the lines 251-253 at the end of the section.

**REPLY:** Keeping in mind that the aim of the PCA was to discriminate lysimeters based on the presence of absence of iron, the sentence was modified as follow (lines 276-278):

"The first two components of the PCA explained 69.4 % of the total variance of the data and discriminated lysimeters depending on the presence or absence of Fe(II) biodissolution in soil waters of the riparian area (Fig. 6)."

L254: In this chapter it may be elaborated on if averaging the lysimeters in time was justified? Does one lysimeter seem to switch from high-nitrate:low-Fe to low-nitrate:high-Fe over time? This is hard to be seen in Fig. 6 as here all lysimeters of one sampling data and one cluster are averaged.

**REPLY:** Please see our previous answer on the PCA approach we used.

Fig. 6: Same as fig. 2 – indicating the different wetness phases would be very helpful here.

**REPLY:** Figures were modified accordingly.

L258 and 270: I don`t understand the idea of a flushing dynamic. Does that mean the solutes are flushed out and replaced by a different water with a different quality? Moreover, I find this to be more an interpretation of that data and thus more a discussion part.

**REPLY:** We agree that this paragraph is unclear and that we used terms more relevant for the discussion. Following this comment, we modified the text as below lines 297-308:

"In cluster 1, DOC, N-NO3 and SRP decreased from 39.8±13.3 to 23.4±8.4 mg L-1, from 2.6±3.6 to 1.2±1.8 mg L-1, and from 0.18±0.18 to 0.08±0.15 mg L-1, respectively, during the rewetting phase of the catchment while Fe(II) was no measured at significant levels. During the high flow period, however, Fe(II) increased gradually from 3.7±3.2 to 26.5±7.8 mg L-1, and both DOC and SRP followed a similar trend with concentrations raising from 27.3±9.5 to 54.9±25.0 mg L-1 and from 0.07±0.13 to 0.18±0.11 mg L-1, respectively. During this period and until the end of the hydrological cycle, N-NO3 were very low, decreasing from 0.54±0.66 mg L-1 at the beginning of the high flow period to values below 0.15 mg L-1 the rest of the survey. The start of the third hydrological period corresponding to the drawdown of the water table and the consecutive aeration of riparian soils was marked by the rapid drop of Fe(II) at 8.1±7.4 mg L-1, DOC at 17.5±10.9 mg L-1, and SRP at 0.02±0.02 mg L-1."

Please note that we applied the same kind of modification for the following paragraph lines 309-321:

"Similarly to cluster 1, soil waters from the cluster 2 exhibited a decline in DOC and SRP concentrations during the rewetting phase of the catchment but these trends continued during the high flow period, with minimal values reached in the middle of February. Thus, DOC dropped from 34.5±7.1 to 9.4±3.1 mg L-1 and SRP from 0.19±0.08 to 0.02±0.01 mg L-1 during this period, before showing an increasing trend to reach concentrations about 21.0±6.1 mg L-1 for DOC and 0.16±0.13 mg L-1 for SRP at the end of the high flow period. DOC remained elevated (24.1±3.1 mg L-1) at the start of the dry period, but SRP dropped close to depletion. In contrast, N-NO3 first increased from 0.57±0.81 mg L-1 in November to maximum values of 6.5±5.9 mg L-1 in the middle of March, and then exhibited decreasing concentrations until a complete depletion at the beginning of the third hydrological period. Contrary to cluster 1, Fe(II) was not measured at significant concentrations in cluster 2 (i.e.

below 0.5 mg L-1) except in March, during which Fe(II) increased from 1.2±1.9 to 4.1±0.2 mg L-1."

*Discussion*

L290: appearance, abundance maybe better than apparition.

**REPLY:** We replaced by the term 'release' that fit more the purpose.

L303: What about a temperature effect here? Or can rainwater just not infiltrate deep enough?

**REPLY:** Although it is true that temperature may affect oxygen solubility and exchanges with atmosphere, we did not found any relationship between Fe(II) and soil temperatures. The gradual increase in Fe(II) also suggests that rainfall events did not impacted iron biodissolution.

L306f: I don't understand the role of the hydraulic gradient here. Translating to soil water/ shallow groundwater travel time? What I read is more about depth to groundwater than hydraulic gradient.

**REPLY:** The hydraulic gradient determines the flow of groundwater coming from upland domain to the riparian soils, impacting therefore the hydrological functioning of valley bottoms (Lambert et al., 2013). Although we are not certain about the exact mechanism involved, we attributed the slight decrease in Fe(II) in February/March to the slight decrease in water table in the upland domain. The paragraph was modified to take into account this comment and the previous one (lines 332-352):

"A fundamental condition for the establishment of reductive conditions is the prolonged waterlogging of riparian soils. As shown earlier for this and other lowland catchments on impervious bedrock, the increase of the hydraulic gradient induced by the rise of groundwater in the upland domain during the high flow period maintains a strong hydrologic connection between upland and riparian domains (Pacific et al., 2010; Molenat et al., 2008). Under these conditions, riparian soils remain waterlogged owing to a high and continuous hillslope groundwater flow, leading to the gradual establishment of reductive conditions and the subsequent triggering of Fe-biodissolution as long as inputs of oxidizing species remained limited and/or counterbalanced by higher rate of consumption through microbial activity (Lotfi-Kalahroodi et al., 2021; Lambert et al., 2013). This pattern is well illustrated by data from lysimeters of the first cluster (Fig. 7). After a quick depletion of an initial stock of nitrate accumulated during the previous summer, reductive conditions were rapidly established at the beginning of the high flow period and increasing Fe(II) concentrations in soil waters lead to the onset of the reductive Fe biodissolution in riparian soils. The gradual increase in Fe(II) during all the high flow period despite variations in temperature or rainfall patterns (with some intense precipitation events > 20 mm d-1) suggests a limited impact of these climatic episodes, except during a period of low precipitation during which both Fe(II) and DOC exhibited a slight decrease in February/March. We attributed this small drop to the drawdown of the water table in upland groundwater flow following a prolonged absence of precipitations (see PK3 fluctuations, Fig. 2) that may have re-oxygenated soil waters (as no changes in N-NO3 occurred)."

L307: If DOC and SRP are similarly affected by oxygen and iron presence, why are they weighted differently in the two clusters?

**REPLY:** Likely because SRP concentrations were lower compared to DOC.

L316: Double word biodissolution here.

**REPLY:** Indeed.

L318f: I am not sure if I understand what is meant by "net depletion pattern".

**REPLY:** Both DOC and SRP concentrations decreased in soil waters, that we interpreted as the progressive flushing of a finite DOC/SRP pool.

L327 (and 313): Spatial patterns are not shown but may be helpful and interesting? So maybe map the clusters back to the catchment figure? Could be in the SI or an additional panel in Fig. 5.

**REPLY:** Lysimeters were aligned along three lines parallel to the stream channel. These lines, about 10-20 m from each other, were located at different distance from the stream with the aim to capture the heterogeneity of water flow paths and nitrates concentrations coming from the upland domain. Despite our sampling design, the distance between each lysimeter is not a variable that we could integrate in our analysis. We would need to set distance to an independent point (the nearest field? the river?) but we don't think this is would lead to an interesting pattern as no spatial pattern was visible: two neighbouring lysimeters could be more different than lysimeters on the opposite side of the

Following this comment, more details were added in the Material and methods (lines 140-144):

"We placed the lysimeters along three lines parallel to the stream channel, about 10-20 m apart from each other and from the stream, with the aim to capture the heterogeneity of water flow paths and nitrate concentration coming from the upslope cultivated fields. Lysimeters were all located in the hydromorphic soils unit according to the soil map (Fig. 1)."

And in Results (lines 260-261):

"Despite the fact that lysimeters were installed along three lines that were more or less closed to the stream channel, no spatial pattern was identified."

L337: Water circulation is maybe not the most precise word here. Does soil water really circulate?

**REPLY:** We modified the sentence line 409:

"A first explanation can be related to the heterogeneity in water flowpaths in soils."

L353-357: This is an interesting part but would, at this process-scale, better fit the end of chapter 4.1? But I am not fully sure either. I noted that this is an initial statement on something explained in more detail below (L373ff). Maybe make more clear that details are given in the following text?

**REPLY:** We think it is a good opening for the 4.2 section. Indeed, the idea behind these sentences (that winter should be considered as an active phase of DOC export) are developed in the next paragraphs where we try to link soil and stream dynamics.

L366-367: I know what you mean here but generally a "supply-limited pool" is associated with a dilution behavior not a flushing behavior. So, check your choice of words here and maybe better describe what concentration dynamics you see at this point in time in the stream.

**REPLY:** We replace by "DOM pool limited in size" which is more relevant.

*Conclusion*

L401: The word "but" indicates some contradiction which I do not clearly see here. Fe-reduction can only establish when nitrates are not present, right?

**REPLY:** Indeed, we should reconsider this sentence. We proposed the following changes lines 496-503:

"In agreement with previous studies (e.g. Selle et al., 2019; Knorr, 2013), the establishment of Fe-reducing conditions in riparian areas was identified as a major mechanism for the release of large amount of DOM in soil waters. In agricultural catchments, however, we found that this process can be buffered by nitrate, leading to a strong spatial heterogeneity in the magnitude of iron biodissolution and its consequences on soil DOC dynamics. Our study also evidenced that another production mechanisms unrelated to Fe dynamics contributed to release DOM in riparian soils during the winter period, pointing to the need to further investigate stream DOC export at the soil/stream interface"

L409: Any reference for the "wetter winter" statement?

**REPLY:** Yes: Strohmenger et al. (2020) has been added.

L404ff: This last section in the conclusions seem to make a new story on the relation of the small-scale redox soil processes to the long-term trends. While I appreciate this part I think this may be already part of the introduction and motivation.

**REPLY:** In fact this was part of the motivation, but indeed this was very briefly mentioned in the manuscript lines 82-85. The last paragraph of the introduction has been modified accordingly (lines 85-95)

"To this end, we installed zero-tension lysimeters in the riparian area of the Kervidy-Naizin catchment, whose stream waters are continuously monitored for water quality, including DOC at high frequency (Fovet et al., 2018). This catchment is located in Brittany (France), a region where stream DOC concentrations exhibited contrasting trends (increasing, decreasing or no trend) over the 2007-2020 period despite similar geomorphological and climatic conditions (Supplementary Fig. S1). The Kervidy-Naizin catchment for instance exhibits a weak but significant increase in stream DOC concentrations over the last two decades (Strohmenger et al., 2020). In this context, another goal of this study was to explore the hypothesis that long-term regional decrease in nitrate inputs (Abbott et al., 2018) have impacted long-term trends in DOC through iron dynamics in riparian soils."

**References**

Dupas, R., Gruau, G., Gu, S., Humbert, G., Jaffrézic, A., and Gascuel-Odoux, C.: Groundwater control of biogeochemical processes causing phosphorus release from riparian wetlands, Water Research, 84, 307-314, https://doi.org/10.1016/j.watres.2015.07.048, 2015.

Lambert, T., Pierson-Wickmann, A.-C., Gruau, G., Jaffrezic, A., Petitjean, P., Thibault, J.-N., and Jeanneau, L.: Hydrologically driven seasonal changes in the sources and production mechanisms of dissolved organic carbon in a small lowland catchment, Water Resources Research, 49, 5792-5803, https://doi.org/10.1002/wrcr.20466, 2013.

Strohmenger, L., Fovet, O., Akkal-Corfini, N., Dupas, R., Durand, P., Faucheux, M., Gruau, G., Hamon, Y., Jaffrezic, A., Minaudo, C., Petitjean, P., and Gascuel-Odoux, C.: Multitemporal Relationships Between the Hydroclimate and Exports of Carbon, Nitrogen, and Phosphorus in a Small Agricultural Watershed, Water Resources Research, 56, e2019WR026323, https://doi.org/10.1029/2019WR026323, 2020.

**Response to Reviewer 3, Benny Selle**

This study reports on measurements of DOC quantity and quality, ferrous Fe, nitrate, P and pH in riparian soils for an agricultural headwater catchment in western France. Zero-tension lysimeters were installed in riparian soils at 15 cm depth. 17 lysimeters were sampled weekly to biweekly between November 2022 and June 2023. From data analysis, reductive Fe dissolution and the associated DOC mobilisation were found to be driven by the availability of nitrate. Redox driven mobilisation of DOC happed during the relatively cool and wet winter months.

General comments:

This is a well written paper that would benefit from the analysis of a few more aspects insufficiently addressed in the manuscript:

**REPLY:** We thank B. Selle for the positive evaluation of our work.

(i) The molar ratio at which Fe and DOC were mobilised should be reported and could be compared to ratios reported in the literature. This would indicate if the processes interpreted from the data are reasonable.

**REPLY:** The mean DOC:Fe(II) molar ratio was 142.4±285.5. This was higher than the DOC:Fe(II) ratio measured in experimental conditions (74.5±74.6) but similar than value measured on the field (134.4±25.6) by Lotfi-Kalahroodi et al. (2021) who aimed to investigate Fe reduction in the riparian area of our study catchment.

To compare our DOC:iron ratio it would have been necessary to measure Fe(III). However, if we consider a ratio between Fe$_{tot}$ and Fe(II) of 4.8 based on Lotfi-Kalahroodi et al. (2021), our DOC:Fe ratio are 29.3±58.8. Keeping in mind that this is a rough estimation, the ratio at which DOC increases in soil waters is consistent with previous studies (e.g. Selle et al., 2019; Musolff et al., 2017; Grybos et al., 2009; Cabezas et al., 2013).

The manuscript has been modified accordingly (lines 358-365):

"Regarding DOC, the mean DOC:Fe(II) molar ratio was 142.4±285.5. This was higher than the DOC:Fe(II) ratio measured in experimental conditions (74.5±74.6) but similar to value measured on the field (134.4±25.6) by Lotfi-Kalahroodi et al. (2021) who aimed to investigate Fe reduction in the riparian area of our study catchment. Fe(III) concentrations in soil waters were not measured, but, based on the work of Lotfi-Kalahroodi et al. (2021), we can estimate a ratio between total Fe and Fe(II) of 4.8. Keeping in mind that this is a rough estimation, our mean DOC:Fe ratio would be about 29.3±58.8, which is consistent with previous studies (e.g. Selle et al., 2019; Musolff et al., 2017; Grybos et al., 2009; Cabezas et al., 2013)."

(ii) Also, there may be an indirect mobilisation of DOC due to a pH increase with Fe reduction which could be evaluated from the data presented. Note that an indirect mobilisation of DOC with a pH increase would probably increase OC to Fe ratios compared to a pure mobilisation due to the dissolution of iron minerals.

**REPLY:** In fact the effect of pH on DOC mobilisation has also been investigated in the study site (Grybos et al., 2009). Results have shown that up to 60% of the release is due to DOC desorption caused by the pH increase that accompanies the reduction of Feoxyhydroxides in these soils. Although pH was variable among lysimeters, there was a positive relationship between DOC and pH (Figure 2), therefore supporting an indirect mobilisation of DOC linked to increasing pH associated with iron reduction.

Discussion has been modified accordingly (lines 365-375):

"The nature of processes releasing DOC upon the reduction of soil-Fe oxyhydroxides in riparian soils of our study site has been studied in laboratory conditions (Grybos et al., 2009). Results have shown that up to 60% of the release is due to DOC desorption caused by the pH increase that accompanies the reduction of Feoxyhydroxides in these soils, the remaining 40% being due to the dissolution of Fe-oxyhydroxides that strongly adsorb organic

compounds previously bounded to surface minerals (e.g. Hagedorn et al., 2000). In good agreement with these results, soil DOC was positively related to pH (Supplementary Fig. S5). The abrupt decrease in DOC in June illustrates the restoration of aerobic conditions owing to the drawdown of the water table in the bottomland domain led to the formation of Fe-minerals and the subsequent retention of DOC and SRP (Gu et al., 2017)."

[Figure]

Figure S5 – Correlation between pH and DOC concentrations for all lysimeters and all dates..

Specific comments:

L39-42: Isn't this a contradiction? DOC export cannot be source and transport limited at the same time.

**REPLY:** This is a matter of time-scale: at the short/daily scale, DOC export is limited by the amount of flow (rain) that circulate within topsoil horizons, but at the annual scale DOC export is limited by the size of the potential pool of mobile DOM. Text has been modified for more clarity (lines 36-40):

"The flushing of shallow organic-rich soil layers during storm events (at the daily scale) typically represents the majority of annual DOC loads (Inamdar et al., 2006), and the DOC versus discharge relationships show that DOC export is transport-limited at the event scale (Buffam et al., 2001; Zarnetske et al., 2018)."

L54-56: This indicates a source limited DOC export, which contradicts the transport limitation stated above. I am confused. Perhaps, the conceptual model of source and transport limitation for DOC export needs to be explained better.

**REPLY:** Considering this comment and the previous one, we realized that indeed the formulation was not easy do understand as the notion of source or transport limited are time-scale dependant. To make our introduction more clear and more focus on the time-scale of the study, we suggest the following changes in the introduction (lines 32-50):

"Numerous research carried out in temperate and boreal regions have shown that headwater catchments are the main entry point of DOM into fluvial networks (Ågren et al., 2007; Creed et al., 2015) and identified riparian areas as the dominant sources of DOM at the catchment scale owing to their location at the terrestrial-aquatic interface (Sanderman et al., 2009; Lambert et al., 2014; Laudon et al., 2012; Winterdahl et al., 2014). The flushing of shallow organic-rich soil layers during storm events typically represents the majority of annual DOC loads (Inamdar et al., 2006), and the DOC versus discharge relationships during storm events show that DOC export is transport-limited at the event scale (Buffam et al., 2001;

Zarnetske et al., 2018). Although geomorphological and climatic conditions regulate DOC loads in aquatic ecosystems (Winterdahl et al., 2014; Laudon et al., 2012), DOC export at the annual scale is commonly conceptualized as a two-steps process in which DOM is produced and stored in the catchment during the hot and dry period, and then exported toward surface waters during the wet and cold period (Boyer et al., 1996). This two-steps conceptual model often described in temperate catchments (Deirmendjian et al., 2018; Strohmenger et al., 2020; Wen et al., 2020; Ruckhaus et al., 2023) is also supported by numerous studies carried out in tropical (Bouillon et al., 2014), boreal (Tiwari et al., 2022), Mediterranean (Butturini and Sabater, 2000) or Arctic fluvial networks (Neff et al., 2006). However the processes regulating the size of the riparian DOM pool remain unclear (Tank et al., 2018 and references below)."

L79: hypothesis instead hypothese

**REPLY:** Correct.

L179: Is DOC part of the mineral composition of soil waters?

**REPLY:** No, this is not. Text has been modified lines 211-213:

"For this reason, data (DOC, NO3-, SRP and Fe(II) concentrations and the relative contribution of PARAFAC components) were averaged for each lysimeters then normalized."

L284: I am not sure if Skerlep et al. is an appropriate reference here: Does this paper report really on Fe reduction during winter periods?

**REPLY:** Indeed, this was a mistake. We added Knorr et al. (2013) and Selle et al. (2019) as more relevant examples.

L316: delete biodissolution

**REPLY:** ok.

L317: delete as long

**REPLY:** ok.

L364-372: Here you again discuss your conceptual model of source versus transport limitations of DOC mobilisation. Perhaps a sketch of the conceptual model would help the reader to better understand this.

**REPLY:** Following a comment from C. Williams, we modified this part of the discussion as our data do not allow us to fully support our statements. Now the paragraph reads as follow (lines 439-458):

"Stable carbon isotopes have indeed demonstrated that riparian soils of the Kervidy-Naizin catchment – and more particularly the DOM-rich uppermost soil horizons – are the dominant source of stream DOC at the catchment scale (Lambert et al., 2014), a feature commonly shared by headwater catchments (e.g. Sanderman et al., 2009). Thus, the decline in DOC and SRP observed in soil waters, particularly in the second cluster whereby these elements became almost depleted (Fig. 7), was consistent with the general flushing behaviour of the catchment shown by stream DOC from November to February. Similarly, the large two to three fold increase in DOC concentrations in riparian soils (in cluster 1 and 2, respectively) denotes a large mobilisation of DOM between March and May despite wet and low temperature conditions, that could explain in turn the pattern observed in stream DOC at the same time. While part of this regeneration can be attributed to iron biodissolution, the release of large amount of DOC the cluster 2 where the reductive biodissolution of Fe(III) was limited implies that another production mechanisms contributed to release DOM in riparian soils. It is

unlikely that agricultural inputs (crop residues, manure application, etc) main may explain the increases in the riparian area, as these sources are episodic and/or size-limited (Lambert et al., 2014; Humbert et al., 2015; Pacific et al., 2010). This observation echoes previous works on the Kervidy-Naizin catchment showing effective inter-annual regeneration mechanisms of the pool of soluble phosphorus in soils unrelated to iron dynamics (Gu et al., 2017), a statement supported here by the fact that SRP concentrations followed a similar pattern as DOC in soils grouped in the second cluster (Fig. 7)."

L384: Do you equate DOC production with redox driven mobilisation of DOC here?

**REPLY:** No, 'several' has been replaced by 'another'.

L386: delete main

**REPLY:** ok.

L409: Trends were not previously discussed but are mentioned now suddenly in the conclusion section.

**REPLY:** The introduction was modify to introduce the long-term patterns identified in Brittany (lines 85-95):

"To this end, we installed zero-tension lysimeters in the riparian area of the Kervidy-Naizin catchment, whose stream waters are continuously monitored for water quality, including DOC at high frequency (Fovet et al., 2018). This catchment is located in Brittany (France), a region where stream DOC concentrations exhibited contrasting trends (increasing, decreasing or no trend) over the 2007-2020 period despite similar geomorphological and climatic conditions (Supplementary Fig. S1). The Kervidy-Naizin catchment for instance exhibits a weak but significant increase in stream DOC concentrations over the last two decades (Strohmenger et al., 2020). In this context, another goal of this study was to explore the hypothesis that long-term regional decrease in nitrate inputs (Abbott et al., 2018) have impacted long-term trends in DOC through iron dynamics in riparian soils."

Figure 4: Why is relation between Fe and DOC is closer than between Fe and P?

**REPLY:** Likely because SRP is also controlled by soil properties such as phosphorus speciation. We modified the text lines 353-357:

"Therefore, large release of DOC occurred in soils of the first cluster. Iron biodissolution also affected SRP, but the relationships was weaker suggesting that the reductive dissolution of soil Fe was not the primary driver of SRP concentrations in soils. For instance, soil properties, and more specifically soil phosphorus content and speciation, have been shown to strongly regulate SRP in soil waters of the Kervidy-Naizin catchment (Gu et al., 2017)."

**Reference**

Cabezas, A., Gelbrecht, J., and Zak, D.: The effect of rewetting drained fens with nitrate-polluted water on dissolved organic carbon and phosphorus release, Ecological Engineering, 53, 79-88, https://doi.org/10.1016/j.ecoleng.2012.12.016, 2013.

Grybos, M., Davranche, M., Gruau, G., Petitjean, P., and Pédrot, M.: Increasing pH drives organic matter solubilization from wetland soils under reducing conditions, Geoderma, 154, 13-19, https://doi.org/10.1016/j.geoderma.2009.09.001, 2009.

Lotfi-Kalahroodi, E., Pierson-Wickmann, A.-C., Rouxel, O., Marsac, R., Bouhnik-Le Coz, M., Hanna, K., and Davranche, M.: More than redox, biological organic ligands control iron isotope fractionation in the riparian wetland, Scientific Reports, 11, 1933, 10.1038/s41598-021-81494-z, 2021.

Musolff, A., Selle, B., Büttner, O., Opitz, M., Knorr, K.-H., Fleckenstein, J. H., Reemtsma, T., and Tittel, J.: Does iron reduction control the release of dissolved organic carbon and phosphate at catchment scales? Need for a joint research effort, Global Change Biology, 23, e5-e6, https://doi.org/10.1111/gcb.13758, 2017.

Selle, B., Knorr, K.-H., and Lischeid, G.: Mobilisation and transport of dissolved organic carbon and iron in peat catchments—Insights from the Lehstenbach stream in Germany using generalised additive models, Hydrological Processes, 33, 3213-3225, https://doi.org/10.1002/hyp.13552, 2019.

---

## Author Response (AR2)

**Public justification (visible to the public if the article is accepted and published)**:
Dear Authors,

Thank you for your careful revision in response to all comments and suggestions of the three reviewers. Two of the same reviewers provided overall positive evaluations of your revised manuscript. Based on their positive evaluations accompanied by a small number of comments, I am pleased to recommend your revised manuscript for acceptance after some technical corrections. Please refer to the following comments offered by the reviewers and additional ones from my own reading:

- Additional comments
Title: Please check which one grammatically and semantically suits better - "Effects (The effect) of A on B" or "Roles (The role) of A in B".
**REPLY:** Thank you for this comment; we changed 'Roles' for 'The Role'.

Line 8: Please include "agricultural" before "headwater catchments".
**REPLY:** Done.

- Reviewer 1
The authors addressed my comments. However, one point struck me right away. The average molar ratio DOC/Fe(II) must be similar to the slope of a regression line through data points presented in Figure 4a. My visual assessment of the slope from the figure would be: (50mgDOC/L-25mgDOC/L)/12mgDOC/mmolDOC)/((25mgFe/L-10mgFe/L)/55.85mgFe/mmolFe)=7.8, which one order of magnitude different to the average ratio of 142 reported in the revised manuscript. Please check if the reported ratio of 142 is correct.
**REPLY:** We verified our calculations, using the same molar ratio as the reviewer, and we can confirm that the mean ratio of 142 is correct. Please note that the standard deviation is high, about 285, reflecting a large variability in DOC:Fe(2) ratio owing to field conditions. Thus, the mean ratio is influenced by high values and this is not reflected in the DOC versus Fe(2) plot.

- Reviewer 2
General comments
The authors did a good job addressing the reviewer's points. There are only very few technical corrections needed from my point of view.
Specific comment
Introduction
L36ff: Check this sentence. „Flushing… represents … loads". It is the process of flushing that is responsible for the majority of exported loads?
**REPLY:** Yes, but for clarity we modified the sentence. We replaced 'Flushing' by 'Subsurface flow through' and 'typically represents' by 'is responsible for'. The sentence now reads:
'Subsurface flow through shallow organic-rich soil layers during storm events (at the daily scale) is responsible for the majority of annual DOC loads […]'

Methods
Figure 1: Consider to place the stream network above the hydromorphic soil layer.
**REPLY:** Done.

L139: Can you check the manuscript for DOM vs. DOC? Not sure if there is a consistent rule when to take what term. Just make sure you have a consistent principle.

**REPLY:** In the manuscript, we used DOM for referring to the dissolved organic material in general (that is all organic molecules, including those that do not contain C) and DOC to specifically refer to the concentration of organic carbon in the filtered samples. DOC is commonly used as a proxy for DOM content. We take care to apply this rule in the whole manuscript.

L208ff: I now better get the point why averaging was done. However, this is not fully clear from the text since you write that you want to cluster to show temporal patterns of each cluster. For this reason you average over time. This is a bit misleading in this order. You may explicitly write that "only" for the sake of clustering a temporal averaging was done. After classification you display and interpret the time series of each cluster. Not sure if that is more clear though…

**REPLY:** We try to make our purpose better in the material and methods: 'For this reason, data (DOC, NO3, SRP and Fe(II) concentrations and the relative contribution of PARAFAC components) were averaged for each lysimeters then normalized **in order to group spatially the lysimeters before investigating temporal patterns**.'

Thank you for submitting your work to Biogeosciences.

Sincerely,

Ji-Hyung Park
Associate Editor, Biogeosciences